# A *Sarcina* bacterium linked to lethal disease in sanctuary chimpanzees in Sierra Leone

Leah A. Owens [1], Barbara Colitti[2], Ismail Hirji [3], Andrea Pizarro[3], Jenny E. Jaffe [4,5], Sophie Moittié [6,7], Kimberly A. Bishop-Lilly[8], Luis A. Estrella[8], Logan J. Voegtly [8,9], Jens H. Kuhn [10], Garret Suen [11], Courtney L. Deblois[11], Christopher D. Dunn[1], Carles Juan-Sallés[12] & Tony L. Goldberg [1✉]

Human and animal infections with bacteria of the genus *Sarcina* (family *Clostridiaceae*) are associated with gastric dilation and emphysematous gastritis. However, the potential roles of sarcinae as commensals or pathogens remain unclear. Here, we investigate a lethal disease of unknown etiology that affects sanctuary chimpanzees (*Pan troglodytes verus*) in Sierra Leone. The disease, which we have named "epizootic neurologic and gastroenteric syndrome" (ENGS), is characterized by neurologic and gastrointestinal signs and results in death of the animals, even after medical treatment. Using a case-control study design, we show that ENGS is strongly associated with *Sarcina* infection. The microorganism is distinct from *Sarcina ventriculi* and other known members of its genus, based on bacterial morphology and growth characteristics. Whole-genome sequencing confirms this distinction and reveals the presence of genetic features that may account for the unusual virulence of the bacterium. Therefore, we propose that this organism be considered the representative of a new species, named "*Candidatus* Sarcina troglodytae". Our results suggest that a heretofore unrecognized complex of related sarcinae likely exists, some of which may be highly virulent. However, the potential role of "*Ca*. S. troglodytae" in the etiology of ENGS, alone or in combination with other factors, remains a topic for future research.

[1] Department of Pathobiological Sciences, School of Veterinary Medicine, University of Wisconsin-Madison, Madison, WI, USA. [2] Department of Veterinary Science, University of Torino, Torino, Italy. [3] Tacugama Chimpanzee Sanctuary, Freetown, Sierra Leone. [4] Tai Chimpanzee Project, Max Planck Institute for Evolutionary Anthropology, Leipzig, Germany. [5] Epidemiology of Highly Pathogenic Microorganisms, Robert Koch Institute, Berlin, Germany. [6] School of Veterinary Medicine and Sciences, University of Nottingham Sutton Bonington Campus, Sutton Bonington, Leicestershire, UK. [7] Twycross Zoo, Atherstone, UK. [8] Genomics and Bioinformatics Department, Biological Defense Research Directorate, Naval Medical Research Center, Fort Detrick, MD, USA. [9] Leidos, Reston, VI, USA. [10] Integrated Research Facility at Fort Detrick, Division of Clinical Research, National Institute of Allergy and Infectious Diseases, National Institutes of Health, Fort Detrick, MD, USA. [11] Department of Bacteriology, University of Wisconsin-Madison, Madison, WI, USA. [12] Noah's Path, Elche, Spain. ✉email: tony.goldberg@wisc.edu

Emerging pathogens pose a substantial risk to animal and human health[1,2]. Pathogens can emerge due to the acquisition of virulence factors through genetic mutation and horizontal gene transfer[3–5] and due to ecological processes that alter their epidemiology/epizootiology and host range[6–8]. These processes are accelerating due to factors such as agricultural intensification[9], demographic shifts[10], ecosystem perturbations[11], geophysical processes[12], and global environmental changes[13]. Furthermore, improved diagnostic methods facilitate detection and characterization of new pathogens and the genetic features that contribute to their emergent phenotypes[14–16].

Emerging pathogens of non-human primates are especially salient examples of this phenomenon because of the high potential for such pathogens to infect humans, who are genetically similar hosts[17,18]. For example, one of the main causative agents of human malaria, *Plasmodium falciparum*, once thought to have co-evolved with humans, actually arose from a recent zoonotic transmission from a western gorilla (*Gorilla gorilla* (Savage, 1847))[19–21]. Nowhere are such risks more evident than in zoological and sanctuary settings, where captive and semi-captive primates come into frequent close contact with people[22,23]. For example, contact with New World primates led to simian foamy virus transmission to primate workers in Brazil[24] and monkeypox virus transmission occurred in staff at a primate sanctuary following a monkeypox outbreak in sanctuary chimpanzees (*Pan troglodytes* (Blumenbach, 1775)) in Cameroon[25].

Since 2005, western chimpanzees (*Pan troglodytes verus* Schwarz, 1934; "chimpanzees" hereafter) in Sierra Leone's Tacugama Chimpanzee Sanctuary (TCS, in Western Area Peninsula National Park) have suffered from a lethal disease of unknown etiology. Characteristic signs are neurologic (weakness, ataxia, seizures) and gastrointestinal (abdominal distension, anorexia, vomiting), resulting in death even after aggressive medical treatment by staff veterinarians. To date, a total of 56 individuals at this facility have died of this condition, which we have named "epizootic neurologic and gastroenteric syndrome" (ENGS), constituting a medical emergency in this population, which averages 93 chimpanzees at any given time. TCS is the largest repository of Sierra Leone's chimpanzee genetic diversity, a training site for conservationists throughout western Africa, an educational/ecotourism destination important for the local economy, and the only home for displaced or orphaned chimpanzees in the country.

Despite ongoing efforts by veterinary staff and international collaborators, the etiology of ENGS has remained elusive. Encephalomyocarditis virus (EMCV; *Picornaviridae*: *Cardiovirus*) infection and toxicity from certain plants (*Dichapetalum toxicarium* (G. Don) Baill./*D. heudelotii* (Planch. ex Oliv.) Baill.) were both suspected but, after investigation, deemed unlikely to be causal. In such circumstances where infection has been suspected but known agents have not been identified, diagnostic approaches based on metagenomics have proven useful[26–28]. We therefore undertook a case-control epizootiological investigation to identify potential pathogens of all major types (viruses, bacteria, and eukaryotes) using metagenomics and traditional methods in the TCS chimpanzees, to detect associations between particular microbes and ENGS.

Here we report the finding of a *Sarcina* genus bacterium in 13 of 19 ENGS cases but no controls. We also report the occurrence of gross and histopathological lesions in affected chimpanzees consistent with the most severe forms of *Sarcina* infection reported in humans and animals. By studying the morphology and growth characteristics of the bacterium, and by sequencing the complete genome of an isolate, we identify features that distinguish it from all previously described members of its genus. In particular, we show that the bacterium possesses genes encoding

biochemical pathways potentially contributing to enhanced virulence, including an encoded urea degradation biochemical pathway, consistent with the clinical signs observed in chimpanzees. We conclude that the genus *Sarcina* likely contains an overlooked complex of species ranging from benign commensals to frank pathogens. In light of these findings, the importance of sarcinae in human and animal clinical disease should be re-evaluated.

## Results

**Epizootiology, clinical signs, and pathology**. From 2005 to 2018, 56 resident chimpanzees of TCS died of ENGS. In 32 of 56 cases, affected individuals displayed signs including anorexia, neuromuscular weakness, ataxia, seizures, vomiting, and abdominal distension (Fig. 1; "Clinical signs" group). Signs persisted for a median of 6 days (range: 1–90 days) prior to recovery or death (Supplementary Table 1). In all recovered cases, clinical disease subsequently recurred and resulted in death. In the remaining 24 cases, individuals were discovered post-mortem with no premonitory signs noted by care staff or developed signs which progressed to death in 12 h or less (Fig. 1; "Sudden death" group). Despite these disparate manifestations, all clinical presentations were associated and clearly recognizable as the same "mystery disease" (described as "unmistakable" by veterinarians). We therefore chose the term "syndrome" to reflect the heterogenous nature of the clinical presentations and the suspicion of a common etiopathogenesis. ENGS represented the highest cause of mortality in this population, affecting 33.7% of chimpanzees and accounting for 63.6% of deaths during this time period (Fig. 2a), with a case fatality rate of 100% and a seasonal distribution peaking in March (Fig. 2b). The etiological agent of ENGS does not appear to be transmitted directly, as there were few instances of cases clustering in time and space (Supplementary Table 1).

Frequent lesions included acute shock (congestion involving multiple organs), neutrophilic margination in the microcirculation, moderate to marked gastric dilation, pulmonary edema, acute aspiration of digestive contents, and acute hemorrhage in the thymus, pancreas, or both. In total, post-mortem evaluation documentation was available for 28 chimpanzees, but in only 17 of these was the gastrointestinal tract assessed. Of these 17 patients, 14 had gross evidence of acute gastric dilation, 1 did not show such evidence, and 2 were inconclusive (Fig. 1). One of the chimpanzees with acute gastric dilation also had massive hemorrhagic diathesis (Fig. 3a) and multiple gas-filled lesions in the cecal wall (emphysematous typhlocolitis; Fig. 3b). Microscopically, these gas-filled lesions were surrounded by infiltrates of macrophages, eosinophils, and multinucleate giant cells (Fig. 3c).

**Samples**. We selected 95 archived samples from 32 chimpanzees for analysis (Supplementary Data 1). These samples comprised 19 individuals (7 males and 12 females) that had died from ENGS (cases), representing a subset of the 56 total cases that occurred since the epizootic began in 2005 (Fig. 2), and 14 healthy individuals (7 males and 7 females) sampled during routine veterinary health checks or, in 2 instances, sampled post-mortem when cause of death was known and clearly unrelated to ENGS (e.g., from trauma; controls). In one instance, matched samples were available from a healthy chimpanzee who subsequently became ill and died from ENGS (hence this individual was first a control and then a case). The chimpanzees in this study ranged in age from 5 to 27 years (median age = 12 years) and were sampled between 14 March 2013 and 11 July 2016, although not all cases within this time period were sampled or available for study. Among ENGS cases included, clinical signs were similar to those

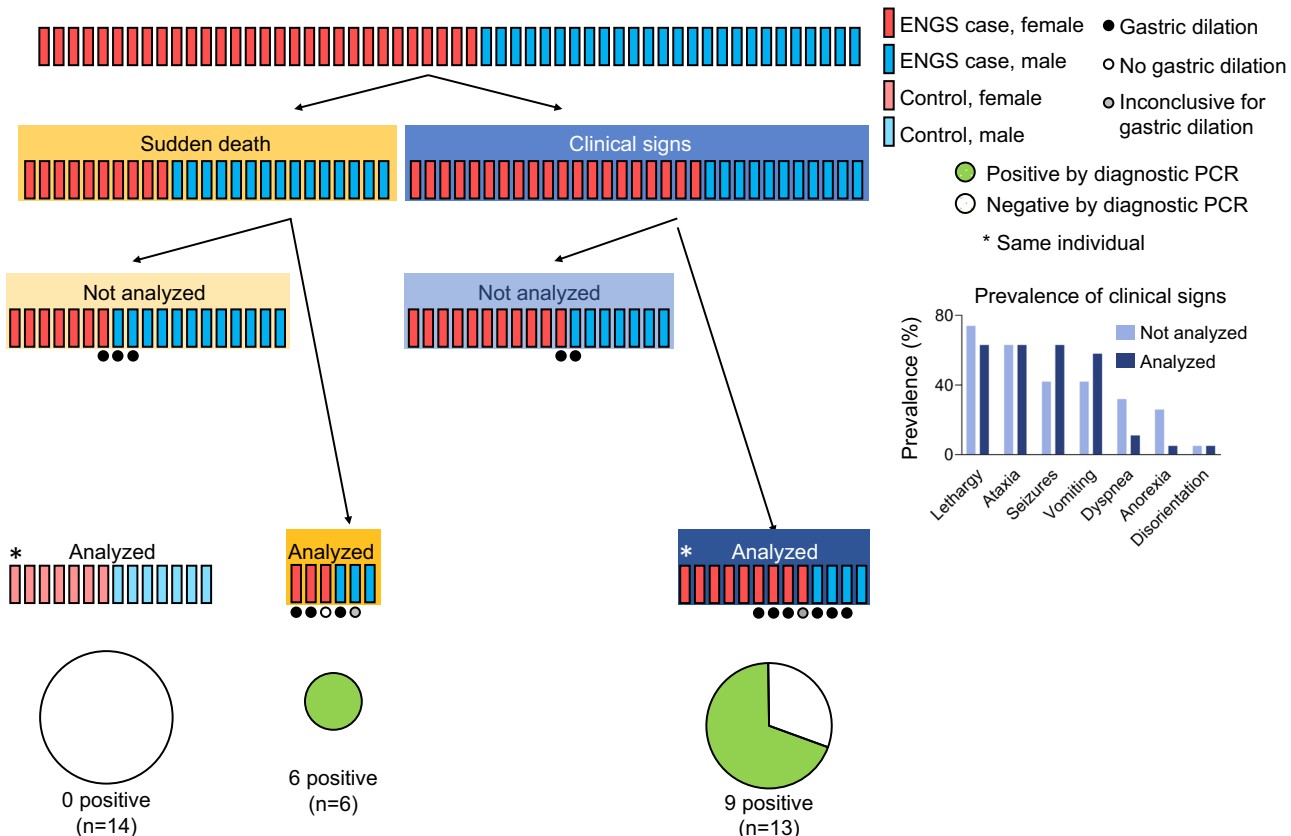

**Fig. 1 Graphic representation of epizootic neurologic and gastroenteric syndrome (ENGS) cases and study samples.** Individual chimpanzees are represented by rectangles and colors denote sex and case/control status (red, female ENGS case, $n = 30$; blue, male ENGS case, $n = 26$; pink, female ENGS control, $n = 7$; light blue, male ENGS control, $n = 7$). Post-mortem findings regarding gastric dilation are shown for those cases where documentation was available ($n = 17$; black circle, gastric dilation, $n = 14$; open circle, no gastric dilation, $n = 1$; filled gray circle, inconclusive for gastric dilation, $n = 2$). "Analyzed" indicates an individual from whom we obtained at least one sample used in this study ($n = 32$) and the asterisk indicates the single individual from whom we obtained samples pre- and post-ENGS. Pie charts are scaled to sample size with green representing individuals positive by diagnostic PCR and white representing individuals negative by diagnostic PCR.

of cases that were not available for inclusion (Fig. 1), the most common of which were ataxia ($n = 12$), seizures ($n = 12$), vomiting ($n = 10$), and abdominal distention ($n = 9$).

**Parasitology.** Microscopic examinations of fecal samples were performed on site for 30 chimpanzees (17 ENGS cases and 13 controls) from 2005 to 2018 using standard direct and sedimentation methods[29], comprising 155 analyses with a median 5 analyses per chimpanzee. Nine records indicated no detectable parasites and 146 records indicated ≥1 parasite, with a median parasite richness of 2. Parasites identified included *Entamoeba* spp., *Troglodytella abrassarti*, *Balantidium coli*, *Ascaris* spp., *Enterobius* spp., *Strongyloides* spp., *Trichostrongylus* spp., *Trichuris* spp., *Taenia* spp., *Schistosoma* spp., and unspecified flagellated protozoa and hookworms (Supplementary Fig. 1), all of which are common in this population of chimpanzees and in apparently healthy animals in other captive and wild settings[29–31]. Fortuitously, three ENGS cases had undergone fecal parasitological examinations immediately prior to or just after the time of death, and only representatives of these same typical/commensal organisms were identified: *T. abrassarti*, *Entamoeba hartmanni*, *B. coli*, and *Enterobius* spp.

Eukaryotic metabarcoding for parasite identification using Earth Microbiome Project (EMP) protocols[32] and previously published primers[33,34] for 12 ENGS cases and 6 controls (24 samples; Supplementary Data 2) yielded a total of 2,955,014 raw reads (1,477,507 paired reads) with good overall sequencing quality (~21% of reads removed during quality filtering; Supplementary Table 2). Due to the pan-eukaryotic nature of the primers, and despite the use of a mammal-blocking primer, the majority of reads were identified as host (~83%; Supplementary Table 2). This identification was not surprising due to the sample types (host tissues) and because of previously published similar findings using the same primers and protocols[35]. We processed data from all samples through all filtering steps, after which we excluded those samples that represented <0.5% of the total filtered data set, resulting in removal of 7 of 24 samples (Supplementary Table 2). From the remaining reads (range 681–31,131 per sample) we identified seven operational taxonomic units (OTUs) representing three parasitic and four environmental eukaryotic organisms (Supplementary Fig. 2). No parasites thus identified were case-associated (i.e. found at statistically significantly different prevalence in case versus control groups using a Fisher's exact test (two-tailed); Table 1).

**Virology.** Metagenomics for virus discovery conducted using previously published methods[36–38] on 12 ENGS cases and 6 controls (24 samples; Supplementary Data 2) generated a total of 151,206,140 reads (mean per sample 6,300,256, standard deviation 3,738,306; average length 161, standard deviation 27). After trimming and filtering on length and quality, 71.3% of sequences remained (mean per sample 4,495,233, standard deviation 2,792,589; average length 113.3, standard deviation 16.3) which were then assembled into 952 contiguous sequences (contigs

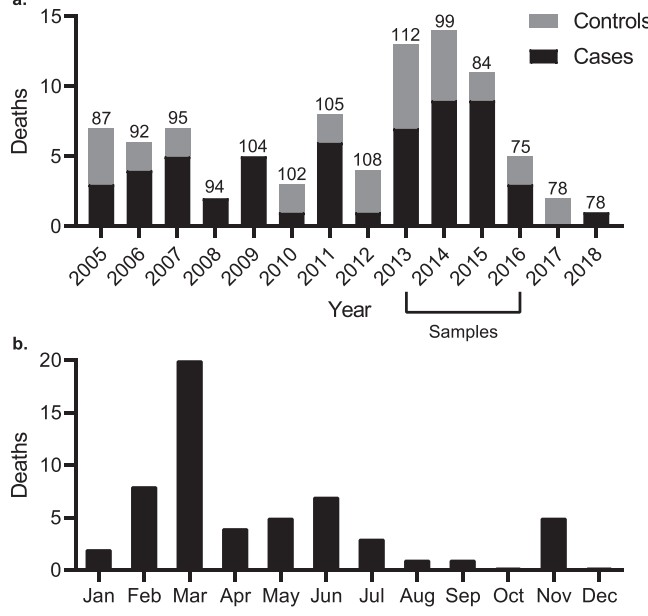

**Fig. 2 Deaths attributed to epizootic neurologic and gastroenteric syndrome (ENGS) at Tacugama Chimpanzee Sanctuary from 2005 through 2018. a** Annual chimpanzee deaths from ENGS (black, overall total = 56) and other causes (gray, overall total = 32). Numbers above the bars indicate the yearly total chimpanzee population and the horizontal bracket below the x-axis denotes the period during which we obtained samples for analysis. **b** Summed totals of ENGS fatalities by month over 2005–2018 (n = 56). Source data are provided as a Source Data file.

hereafter; mean per sample 39.7, standard deviation 45.5; average length per contig 969.5, standard deviation 690.8) at an average sequence depth of 63.4 (standard deviation 172.4, minimum 3.5, maximum 851). Overall, 21.0% of reads assembled into contigs and 79.0% did not. Analyses of sequence data at the individual read level confirmed the results of the analysis of contigs (i.e. identified the same viruses) and did not identify any additional viruses. Eleven viruses were thus identified, each of which was identical to or closely related to a known virus (Supplementary Table 3), and no viruses were case-associated (Table 1). One control animal was infected with a rhinovirus C (*Picornaviridae*: *Enterovirus*) subsequent to an outbreak of respiratory illness in the population. This pathogen was previously documented as a cause of epizootic respiratory disease in wild chimpanzees[37], but the clinical features of this rhinovirus C infection (characterized by upper respiratory signs) are not consistent with ENGS.

**Bacterial metabarcoding.** PCR amplification of the 16S rDNA V4 region was attempted on all 96 samples (19 cases and 14 controls; Supplementary Data 1) which yielded amplicons in 10 of 19 ENGS cases (35 total samples) but none of the controls (0 samples). In total, 1,131,561 raw sequences were generated for all 35 samples of which 787,263 were high quality after filtering in mothur. Good's coverage estimation was calculated for all samples and only those samples that had a value >0.99 were retained for downstream analysis. As a result, 23 samples were considered for this analysis (Supplementary Data 2), which totaled 774,339 high-quality sequences with an average of 33,667 ± 17,122 standard deviation per sample. These sequences were binned at 97% similarity into 2592 OTUs. The OTU counts were normalized to 5900 sequences per sample, and these normalized sequences were used for all further analyses.

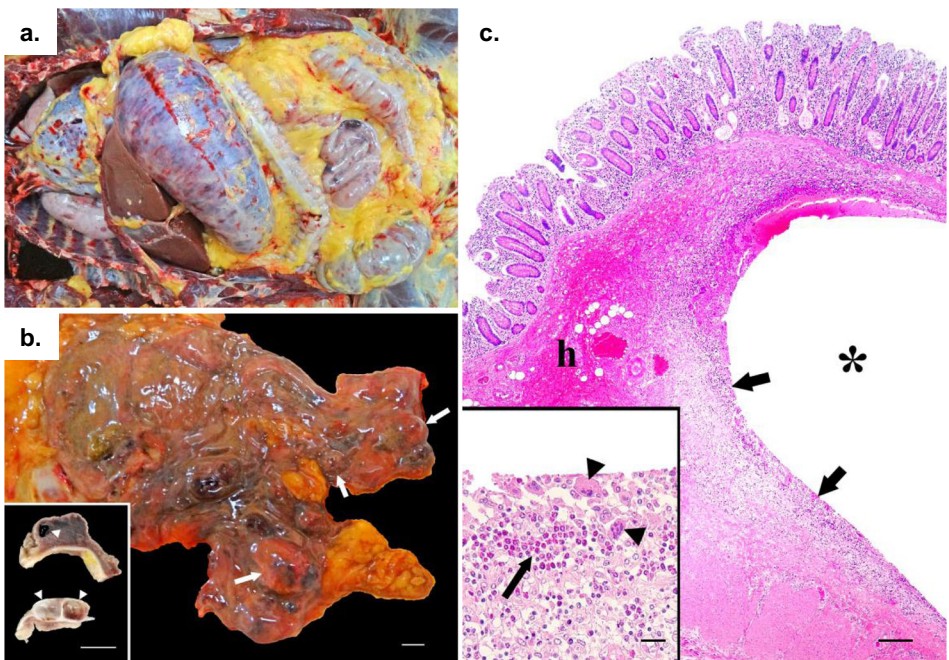

**Fig. 3 Gross and histopathologic images of a chimpanzee that died of epizootic neurologic and gastroenteric syndrome (ENGS).** Photographic images from an adult male chimpanzee who died of ENGS showing moderate to severe gastric dilation (**a**), hemorrhagic diathesis (**a**), and emphysematous typhlocolitis (**b**; arrows point to gas-filled pockets within the cecum wall, which has reddened areas [in inset, white arrowheads point to gas bubbles in the cut surfaces of the formalin-fixed cecum]; scale bars = 1 cm). On histology (**c**), gas-filled space (*) in the cecal submucosa, surrounded by inflammatory infiltrates (arrows) and hemorrhage (h) were visualized by hematoxylin and eosin staining (inset depicts inflammatory infiltrates, which include eosinophils [arrow] and multinucleate giant cells [arrowheads]; scale bars = 300 μm [main image] and 30 μm [inset]; micrograph is a representative image of three similar sections obtained from the same individual).

**Table 1 Case/Control statistical summary.**

| Organism ID | Organism type | Diagnostic | Prevalence in cases (%) | Prevalence in controls (%) | $P^a$ | Odds ratio | 95% CI low | 95% CI high |
|---|---|---|---|---|---|---|---|---|
| *Blastocystis* | Parasite | 18S MB | 50.00 | 33.33 | 0.633 | 2.000 | 0.244 | 16.363 |
| *Oesophagostomum* | Parasite | 18S MB | 0.00 | 16.67 | 0.375 | 0.175 | 0.006 | 5.041 |
| *Troglodytella* | Parasite | 18S MB | 30.00 | 33.33 | >0.999 | 0.857 | 0.098 | 7.510 |
| GB virus C | Virus | Virus Seq | 70.59 | 100.00 | 0.273 | 0.175 | 0.008 | 3.678 |
| Rhinovirus C | Virus | Virus Seq | 0.00 | 16.67 | 0.261 | 0.105 | 0.004 | 2.959 |
| Gemykibivirus 2 | Virus | Virus Seq | 41.18 | 83.33 | 0.155 | 0.140 | 0.013 | 1.474 |
| Chimpanzee parvovirus | Virus | Virus Seq | 5.88 | 0.00 | >0.999 | 1.182 | 0.042 | 32.915 |
| Human picobirnavirus 4 | Virus | Virus Seq | 23.53 | 0.00 | 0.309 | 4.333 | 0.202 | 93.159 |
| Macaque picobirnavirus 24 | Virus | Virus Seq | 11.76 | 16.67 | >0.999 | 0.667 | 0.049 | 9.022 |
| Chimpanzee anellovirus | Virus | Virus Seq | 11.76 | 50.00 | 0.089 | 0.133 | 0.015 | 1.176 |
| Torque teno virus 4 | Virus | Virus Seq | 23.53 | 16.67 | >0.999 | 1.538 | 0.137 | 17.335 |
| Torque teno virus 23 | Virus | Virus Seq | 35.29 | 33.33 | >0.999 | 1.091 | 0.153 | 7.802 |
| Torque teno virus 14 | Virus | Virus Seq | 23.53 | 33.33 | >0.999 | 0.615 | 0.081 | 4.704 |
| Torque teno virus 16 | Virus | Virus Seq | 17.65 | 33.33 | 0.576 | 0.429 | 0.052 | 3.522 |
| "Ca. S. troglodytae" | Bacterium | PCR | 68.42 | 0.00 | 0.0001 | 56.077 | 2.866 | 1097.182 |

*Virus Seq* Virome shotgun sequencing, *18S MB* 18S metabarcoding, *CI* Confidence interval around odds ratio.
ªFisher's Exact test, 2-tailed.

Analysis of these samples showed that 9 of 23 samples contained a large proportion of sequences (>5% of total reads and up to 97.4% in one sample) belonging to a single OTU belonging to an unknown member of the bacterial family *Clostridiaceae* (Clostridia: Clostridiales) (Fig. 4a). This OTU most closely resembled *Clostridium perfringens* in the Greengenes database[39,40]. However, *C. perfringens* diagnostic PCR using published protocols[41–43] failed to yield amplicons in any instances, including in tissues found positive by 16S sequencing.

Re-examination of the representative sequence from this OTU against the National Center for Biotechnology Information's (NCBI's) GenBank (GenBank hereafter) non-redundant database excluding uncultured organisms identified a putative match (97.2% nucleotide identity) to *Clostridium (Sarcina) ventriculi* from feces of Japanese macaques (*Macaca fuscata* Blyth, 1875; GenBank accession numbers LC101491 and LC101492). Re-examination of all samples by including the *Clostridium (Sarcina) ventriculi* sequence from GenBank in the Greengenes database demonstrated this organism to be present in all 23 ENGS case samples (Supplementary Data 2). Notably, the organism was present not only in gastrointestinal contents but also in internal organs such as brain, liver, and spleen, sometimes at high abundance (Fig. 4b). The nomenclature of the genus *Sarcina* is contested (and sometimes the genus name *Clostridium* is substituted) because *Sarcina* is phylogenetically situated within the "cluster I" group of *Clostridia*[44], considered the "true" *Clostridia*, although these organisms are polyphyletic. A proposal was made to change the name *Sarcina* to *Clostridium*, but was not approved because the name *Sarcina* predates the name *Clostridium* and therefore has priority[45].

**Diagnostic PCR**. Oligonucleotide primers specific to the 16S rDNA gene of the unknown organism were successfully developed. PCR with these primers yielded amplicons of the predicted length (289 base pairs [bp]) in 13 of 19 ENGS cases (68.4%) but 0 of 13 controls (Supplementary Fig. 3) which was statistically significant (odds ratio = 56.1; 95% CI 2.87–1097.2; Fisher's exact $P = 0.0001$, two-tailed; Table 1). For one individual, blood samples were available both before and after clinical illness and death

from ENGS; the pre-illness blood sample (collected in February 2016) was PCR-negative whereas the post-mortem blood sample (collected in July 2016) was PCR-positive. Sanger sequences of all amplicons were identical to each other and to the representative sequence generated by metabarcoding, except for one sample with a single nucleotide polymorphism ($C \rightarrow T$ transition) at position 51 of the diagnostic fragment. Sequences of the diagnostic fragment were also identical to published 16S rDNA sequences in GenBank for *S. ventriculi* (AF110272) and *Clostridium ventriculi* DSM286 (NR026146), with the exception of the one variant sequence (1 nucleotide mismatch to the aforementioned published sequences).

**Bacterial isolation and characterization**. We attempted to culture the bacterium using 44 combinations of cell preparations and culture conditions (Supplementary Table 4), 2 of which resulted in growth of colonies that resembled sarcinae. Specifically, wet mounts of colonies grown on egg yolk agar plates and *Sarcina ventriculi* growth medium (SVGM) plates revealed refractile, cuboid cells in packets, a morphology that is distinctive of members of the genus *Sarcina* (Fig. 5a, right panel). These colonies were derived from the liver of one individual (1 colony on egg yolk agar plates) and the brain of another (many colonies on SVGM plates; Supplementary Data 2). We confirmed the identity of every colony using diagnostic PCR and Sanger sequencing (see above).

We were repeatedly able to isolate the organism by plating brain tissue onto SVGM plates, but the organism ceased to remain viable after 2–3 passages and did not grow in any of the seven liquid media tested (Supplementary Table 4). These results are consistent with previous studies reporting great difficulty in isolating and propagating sarcinae[46,47]. Furthermore, we were unable to recover live organisms after freezing colonies placed in 10% or 20% glycerol under various conditions. In contrast, we successfully grew the type strain *S. ventriculi* "Goodsir" (American Type Culture Collection [ATCC] 29068) under the same conditions with ease, including propagating the strain in solid and liquid media, freezing the strain in 10% glycerol, and subsequently recovering the bacterium. On SVGM media, our

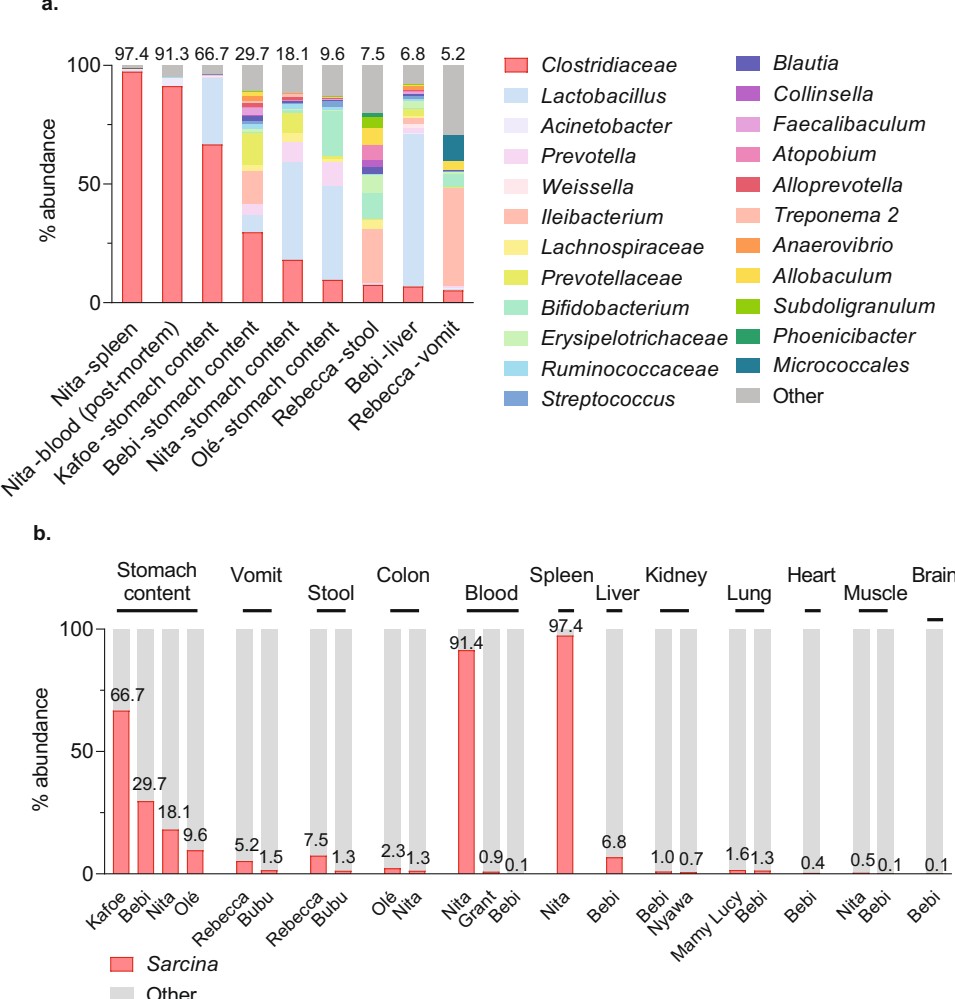

**Fig. 4 Bacterial 16S rDNA microbiome analysis of epizootic neurologic and gastroenteric syndrome (ENGS) case samples. a** The percentage of total reads ($n = 5900$ per sample) from 9 ENGS case samples at genus-level OTU with % reads mapped to the *Clostridiacea* OTU shown on top of each bar. **b** Percent abundance of reads from 23 ENGS case samples classified to genus-level OTUs *Sarcina* (% above each bar) and other, arranged by tissue type.

isolate (JB1) grew more slowly than *S. ventriculi* "Goodsir" (~3–4 days until colonies were visible for JB1, versus 24 h for *S. ventriculi* "Goodsir"). The JB1 isolate displayed morphology (Fig. 5a), Gram's staining characteristics (Fig. 5b), and methylene blue staining characteristics (Fig. 5c) similar to *S. ventriculi* "Goodsir", as both were Gram-positive with a darkly staining outer layer. However, JB1 cells were statistically significantly larger than those of *S. ventriculi* "Goodsir" (mean diameters of 4.29 μm versus 2.83 μm, respectively, Mann–Whitney $U$ $P = 0.0006$, two-tailed; Fig. 5d). The cellular diameter of JB1 falls within the published range for *S. maxima* (4–4.5 μm)[48], but methylene blue staining showed a cellulose-containing cell wall for isolate JB1 which is not characteristic of *S. maxima* (Fig. 5c). In addition, the flattened cellular morphology and large packet size of JB1 cells resemble *S. ventriculi* and not *S. maxima*[49–51].

Archived histologic preparations of tissues collected from ENGS cases during postmortem examination and stained with hematoxylin and eosin clearly revealed sarcinae, visible as packets of darkly staining basophilic cells in gastric contents of the chimpanzee with hemorrhagic diathesis, gastric dilation, and emphysematous gastritis, and in the pulmonary alveoli of another chimpanzee (Fig. 6a). A wet mount direct smear of homogenized brain tissue from the aforementioned ENGS case also demonstrated the presence of packets of sarcinae (Fig. 6b).

**16S rDNA phylogeny**. Alignment of 16S rDNA sequences from the "cluster I" group of *Clostridia*[44], including the new organism (isolate JB1), yielded a final alignment length of 1585 positions. A maximum-likelihood phylogeny built from this alignment shows isolate JB1 to represent a sister taxon to *S. ventriculi*, forming a clade with *S. maxima*, *Eubacterium tarantellae*, and *C. perfringens* (Fig. 7). Bacteria of 13 other recognized species pairs included in the analysis had a lower phylogenetic distance between them than the distance between isolate JB1 and *S. ventriculi* (Supplementary Table 5), lending support to the designation of the bacterium as a representative of a distinct species. To reflect the discovery of this bacterium in chimpanzees (*Pan troglodytes* spp.), we designated it "*Candidatus* S. troglodytae". We propose the *Candidatus* designation in this instance because we were unable to generate a culture suitable for deposition in the requisite two publicly accessible culture repositories in two different countries[52].

**Whole-genome sequencing, assembly, and annotation**. To generate sufficient material for whole-genome sequencing, we repeated bacterial isolation from the brain tissue described above using identical methods and allowed colonies to grow to a large size on SVGM plates. We then harvested a single, large colony, confirmed its identity using microscopy and PCR/sequencing,

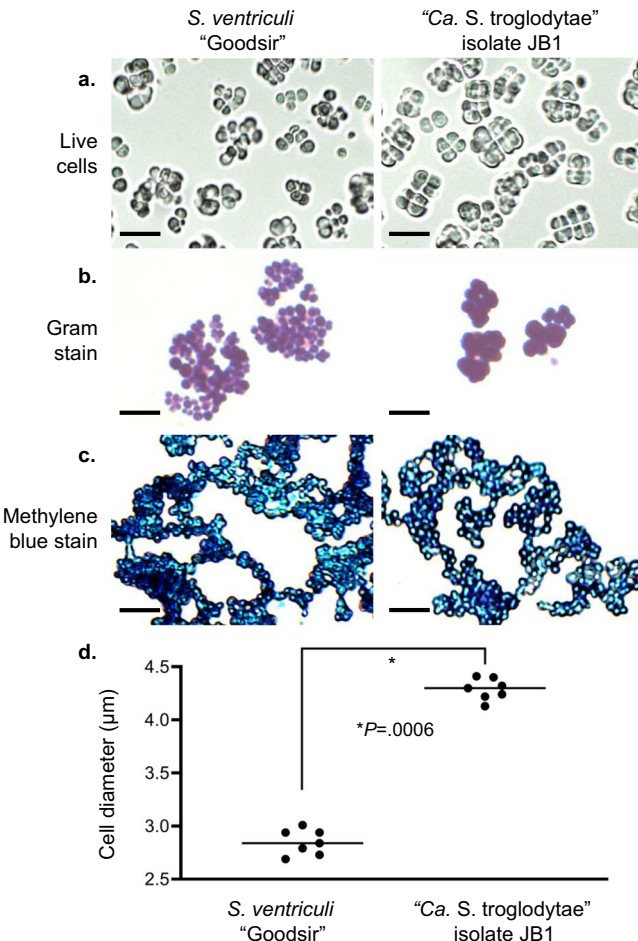

**Fig. 5 Comparative morphology of type strain *Sarcina ventriculi* "Goodsir" ATCC 29068 and "*Ca*. S. troglodytae" isolate JB1. a–c** Cells were imaged live (**a**) or heat-fixed and stained with Gram stain (**b**) or methylene blue stain (**c**). Micrographs are from a single experiment and are representative of three independent experiments with similar results. Scale bars = 10 μm. **d** Live cell diameters were compared, with lines representing median diameters among seven distinct bacterial colonies. Source data are provided as a Source Data file. *Calculated using a Mann–Whitney *U* test, two-tailed.

enzymes, including the urease sub-units alpha, beta, and gamma, and ORFs whose products are predicted to be urease accessory proteins UreE, UreF, and UreG, which are needed for urease maturation. Ureases are nickel-containing enzymes, found in a variety of bacteria, which catalyze the breakdown of urea (a ubiquitous metabolic byproduct in most animals) to ammonia and carbon dioxide[57]. Bacterial and fungal ureases play a key role in gastrointestinal tract colonization and in chronic human diseases such as gastritis and peptic ulcers[58]. Additionally, in the extrachromosomal sequences, we found a 34 kb plasmidial prophage containing ORFs for an ATP-binding cassette (ABC) transporter. ABC transporters use ATP to move specific substrates across a cellular membrane and can function either as an importer (e.g., for uptake of nutrients) or an exporter (e.g., to efflux toxic molecules, including xenobiotic compounds such as drugs)[59]. ABC importers have been associated with increased bacterial survival during colonization of hosts[60] and exporters with bacterial drug-resistance[61], both of which may enhance pathogenesis of an organism.

Sarcinae are not known to produce toxins[62]. However, because of the unusual neurologic disease associated with ENGS, we scanned the genome sequence of "*Ca*. S. troglodytae" for ORFs with sequence homology to known virulence genes using ShortBRED[63] and a customized version of the Virulence Factor Database (VFDB)[64], but found no evidence of such genes. Using the Comprehensive Antibiotic Resistance Database (CARD)[65], we identified two antibiotic resistance ORFs, *OXA-241* on the chromosome and *salA* on plasmid 1, which confer resistance to carbapenems (*OXA-241*) and lincosamides and streptogramins (*salA*)[66,67].

**Summary description of the provisional species**. "*Candidatus* Sarcina troglodytae" is a proposed member of the established genus *Sarcina*, most closely related to *S. ventriculi*, as determined by full-length 16S rDNA phylogenetic analysis. It is an uncultivated, Gram-positive coccus with a tetrad structure and slightly flattened cell morphology and may be identified using the PCR primers: Tacu-Sarc_Diag_F: 5′-TGAAAGGCATCTTTTAACAATCAAAG-3′ and TacuSarc_Diag_R: 5′-TACCGTCATTATCGTCCCTAAA-3′ or the full genome sequence (accessions CP051754–CP051764). We isolated "*Ca*. S. troglodytae" in an anaerobic environment and at mesophilic temperature (37 °C) but were unable to maintain a viable culture for deposition in at least two publicly accessible culture collections, hence the *Candidatus* status. Samples described here are derived from the brain, liver, and lung tissues of sanctuary western chimpanzees (*Pan troglodytes verus*) diagnosed with ENGS.

## Discussion

The genus *Sarcina* within the *Clostridiaceae* is poorly studied in comparison to the highly studied toxigenic clostridia. In 1842, Goodsir described the type species, *S. ventriculi*, in the stomach contents of a human patient with recurrent vomiting[68]. Subsequent studies have provided evidence that bacteria morphologically consistent with *S. ventriculi* cause abdominal pain, nausea, anorexia, vomiting, hematemesis, dysphagia, diarrhea, and generalized weakness in people[69], with esophagitis[70] and duodenitis[71] noted surgically or as a post-mortem finding[72]. Morphologically indistinguishable bacteria assumed to be *S. ventriculi* have also been associated with abomasal bloat in young pre-ruminant animals[73–75], characterized by sudden onset of anorexia, abdominal discomfort, lethargy, dehydration, and shock culminating in high lethality (75–100%) despite treatment[76]. Gastric dilation in monogastric animals (horses, dogs, and cats) has also been linked to putative *S. ventriculi* infection[77,78].

and extracted DNA from this colony (isolate JB2). We performed whole-genome sequencing using a hybrid approach of mate-pair and shotgun sequencing (Supplementary Table 6) followed by de novo genomic assembly using SPAdes[53] and in silico genome closure. The resulting full, high-quality "*Ca*. S. troglodytae" assembly consists of a circular chromosome of 2,435,860 base pairs resolved into one single contig and 10 plasmids (totaling 205,993 base pairs, range: 4.6–78.9 kb; Supplementary Table 7).

We annotated the genome with PATRIC[54] and confirmed that the organism is closely related, but not identical, to *S. ventriculi* (98.5% nucleotide similarity; Supplementary Table 8)[55,56]. Total GC content in our organism was 27.6%, which is similar to that of *S. ventriculi* (27.7%). The organism shares 96.5% of its open-reading frames (ORFs) with *S. ventriculi*, with notable differences in sugar pathways and capsule biosynthesis (Fig. 8). The genome of the JB2 strain contains DNA elements encoding metabolic pathways with the potential for formation of bacterial endospores in addition to anaerobic fermentation pathways, including alcohol fermentation, sulfur reduction, and nitrogen reduction. Interestingly, the "*Ca*. S. troglodytae" genome, but not *S. ventriculi*, possesses ORFs encoding for urea degradation

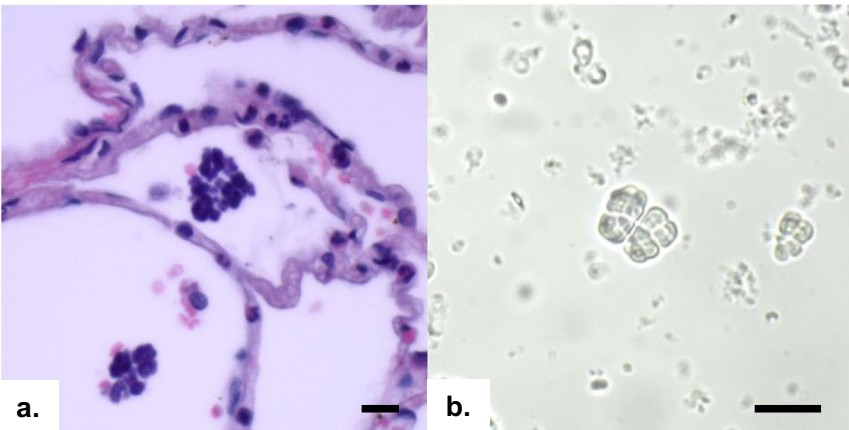

**Fig. 6 Characteristic cuboidal packets of *Sarcina*-like organisms in tissues of ENGS-affected chimpanzees. a** Basophilic packets of cells in a tetrad formation can be seen amongst and within alveoli of hematoxylin and eosin-stained lung tissue of one individual ("Jumu"). **b** Unstained brain tissue homogenate from another individual ("Joko") contains highly refractile, cuboid packets of cells. Scale bars = 10 μm. Micrographs are representative images of one of at least three sections (**a**) or smears (**b**) obtained from the same individual with similar results.

Our results demonstrate a statistically significant association between a bacterium, "*Ca.* S. troglodytae", and ENGS, a protracted lethal epizootic syndrome in sanctuary chimpanzees in Sierra Leone. We designed a case-control epizootiological study using a case definition that encompassed the range of clinical presentations associated with the syndrome, which included both sudden death and gastrointestinal and neurologic signs prior to death. This case definition will likely become more refined as the syndrome is studied further. By applying metabarcoding and metagenomics, we did not find differences between case and control groups with respect to infection with any parasite or virus and therefore deemed these types of organisms unlikely to be causes of ENGS. However, bacterial metabarcoding and a subsequent PCR revealed infection with "*Ca.* S. troglodytae" in 68.4% of ENGS cases but no controls. In one instance, a chimpanzee was PCR-negative for "*Ca.* S. troglodytae" when healthy but subsequently became PCR-positive after succumbing to ENGS.

Sarcinae are notoriously difficult to culture, particularly from non-environmental sources such as animal tissues[46,47]. Despite being studied since the 1800s, sarcinae have been isolated successfully from only a handful of animal or human sources[75,79,80] (see Supplementary Data 3 for review). Prior to this study, only one photomicrograph of unfixed sarcinae cells in their native morphology was published[81]. Although we were able to isolate JB1, it did not survive repeated passages or freezing, distinguishing it from its closest relative, *S. ventriculi*, as does its larger cell size in culture and slower growth. Flattened cell morphology and a cellulose-containing cell wall distinguish "*Ca.* S. troglodytae" from *S. maxima*[51], its next closest relative, despite overlapping cell size. Phylogenetic analysis based on 16S rDNA demonstrates the difference between "*Ca.* S. troglodytae" and *S. ventriculi* to be greater than the difference between bacteria of 13 other recognized species pairs within the clostridial rDNA group I[82]. Whole-genome sequencing and genetic characterization revealed 69 ORFs that were not found in the genome of *S. ventriculi* "Goodsir." For these reasons, we propose that this organism be considered the representative of a new species within the genus *Sarcina*.

Bacteria within the family *Clostridiaceae* include organisms linked to life-threatening diseases, as well as benign commensals and environmental bacteria[62]. Patterns of virulence/toxigenicity do not correspond to phylogeny, and pathogenicity cannot be predicted based on 16S rDNA sequence grouping alone[83]. Whole genome sequencing of "*Ca.* S. troglodytae" revealed no toxin ORFs similar to those present in toxigenic clostridia[84]. The pathogenic effects of "*Ca.* S. troglodytae" on chimpanzees may therefore be caused by mechanisms other than toxicity. For example sarcinae have an unusual yeast-like metabolism[85] that is active over a wide pH range[86], allowing bacteria to produce carbon dioxide gas and ethanol prolifically, both of which can cause disease in the gastrointestinal tract and the central nervous system[87,88].

We also found that the genome sequence of "*Ca.* S. troglodytae" contains ORFs encoding for a predicted urease. Although urease expression is associated with normal microbial flora in some instances, ureases are better known as a key virulence factors in pathogenic bacteria such as *C. perfringens*, *Helicobacter pylori*, and *Klebsiella pneumoniae*[89] and are associated with diseases including ammonia encephalopathy, hepatic encephalopathy, hepatic coma, and gastroduodenal infections[57]. Because, in other bacteria, ureases have established roles in infection and persistence in the host[90], stimulation of host inflammatory reactions[91], cytotoxic effects on host cells[92], and damage to extracellular matrix[93] and tight junctions[94], the presence of a urease biochemical pathway in "*Ca.* S. troglodytae" could help explain the bacterium's pathogenesis and dissemination outside of the gastrointestinal tract. For example, urease activity in the yeast *Cryptococcus neoformans* is responsible for central nervous system invasion; unlike the wild-type organism, mutants lacking this enzyme do not disseminate to the brain and cause meningoencephalitis[95]. Moreover, the major product of urea degradation, ammonia, could enhance the ability of "*Ca.* S. troglodytae" to cause neurologic signs, because ammonia is highly neurocytotoxic in vivo[96].

In some cases of *Sarcina* infection in humans, symptoms are preceded by evidence of delayed gastric emptying[69,70,97–99]. With ENGS, however, affected chimpanzees appeared healthy prior to the onset of signs. It is therefore noteworthy that several studies have shown colonization of *Sarcina* and lesions in the absence of delayed gastric emptying[71,100,101]. For example, a recent publication concerning a lethal case of human emphysematous gastritis highlights several similarities to ENGS, including lack of gastroparesis, afebrile and normotensive presentation, gastrointestinal and neurologic signs, and rapid death[102]. The occurrence of acute gastric dilation and emphysematous lesions in the digestive tract of one ENGS case included in our study recalls cases of *Sarcina* infection in humans and other animal species, which include acute gastric dilation and emphysematous gastritis.

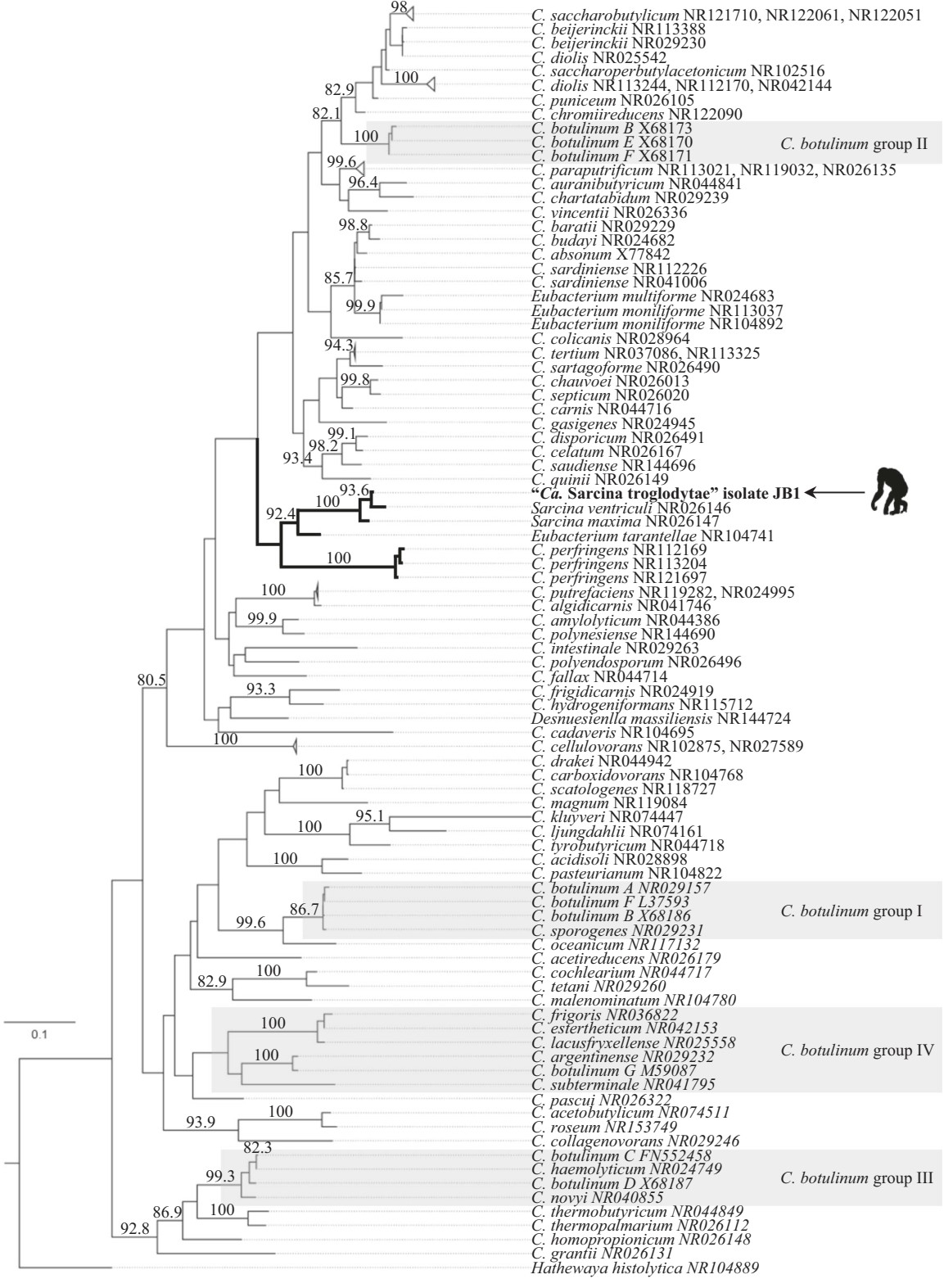

**Fig. 7 Maximum-likelihood 16S rDNA gene phylogeny of the *Clostridiaceae*.** The phylogeny is based on the complete 16S rDNA sequence of "*Ca.* Sarcina troglodytae" isolate JB1 (arrow and silhouette) and 98 other *Clostridia sensu stricto*, with *Hathewaya histolytica* as the outgroup. Gray boxes indicate *Clostridium botulinum* groups[62]. Numbers above the branches are bootstrap values (%) based on 1000 bootstrap replicates (only values ≥75% are shown). Scale bar indicates nucleotide substitutions per site.

A more consistent gross and histopathologic evaluation of affected chimpanzees in the future may reveal a higher proportion of affected chimpanzees because acute gastric dilation and emphysematous gastrointestinal lesions may be misinterpreted as autolysis or overlooked grossly. For example, a chimpanzee in this population who died of ENGS subsequent to the analyses presented here clearly showed emphysematous lesions throughout the gastrointestinal tract. Although each of the clinical

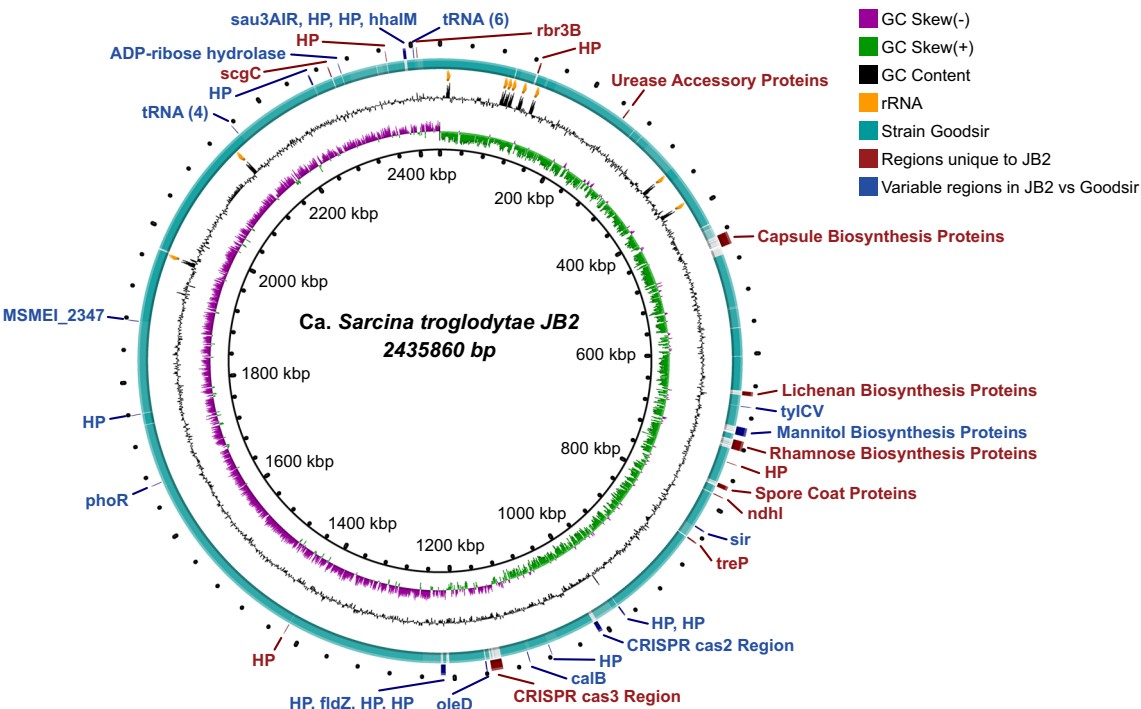

**Fig. 8 Whole chromosome comparison of "*Ca.* Sarcina troglodytae" isolate JB2 compared to the type strain *S. ventriculi* "Goodsir".** Red: regions which are unique to JB2. Blue: different ORF sets in the same relative region in JB2 and Goodsir. Teal: 90–100% identity gradient between JB2 and Goodsir (Goodsir chromosome sequence is based on a manually scaffolded genome).

characteristics of ENGS (abdominal distention, nausea, vomiting, anorexia, diarrhea, and neurologic deficits) are common to diverse diseases, they are all consistent with emphysematous gastroenteritis, as is acute lethality, which likely results from irreversible hemodynamic instability and resulting systemic shock in emphysematous gastritis cases[103].

Notably, we found "*Ca.* S. troglodytae" not only in the gastrointestinal tracts of affected individuals, but also in internal organs, including the brain. For DNA extraction and culture, all tissues were maintained on dry ice, carefully sectioned using sterile technique, and subsampled from the innermost area while still frozen, leading us to conclude that our findings are not likely due to environmental contamination and instead reflect true infection. In human cases of *S. ventriculi* infection, there is precedent for bacteremia, likely originating from gastrointestinal translocation[104,105], but to our knowledge presence of viable sarcinae in the central nervous system has not been previously reported. Central nervous system colonization may therefore be an overlooked clinical feature of severe *Sarcina* infection, or "*Ca.* S. troglodytae" may be a particularly virulent bacterium within the genus *Sarcina*. We advocate that prior and future human and animal cases of severe disease associated with sarcinae, particularly those cases without clear predisposing factors, be revisited, as they could represent other heretofore unrecognized presentations of infection with *Sarcina* bacteria. We also speculate that many documented cases of infection that were assumed to be caused by *S. ventriculi* based on morphology alone may actually have been caused by infection with taxonomically distinct sarcinae. If so, the genus *Sarcina* may contain a complex of morphologically cryptic species varying from benign environmental bacteria to lethal pathogens.

Many questions regarding ENGS and "*Ca.* S. troglodytae" remain unexplained. For example, epizootiologically, ENGS incidence peaks in March each year. As with other disease-associated clostridia, sarcinae form environmentally stable spores[106] and may be ubiquitous in soil[79,99,107], but environmental factors may

contribute to germination of spores and overgrowth. Seasonal changes known to be important in chimpanzees, such as habitat use, diet (including exposure to plant or arthropod toxins), and physiological condition[108], may increase infection risk at certain times of the year. Alternatively, "*Ca.* S. troglodytae" may be reintroduced seasonally, for instance by migratory animals[109], perhaps with seasonal weather patterns facilitating its establishment[110]. The potential role of "*Ca.* S. troglodytae" in the etiology of ENGS, alone or in combination with other factors, remains a topic for future research.

Due to lack of infrastructure at TCS and limitations on sample shipments to the U.S., our analysis could only include samples obtained between 2013 and 2016, even though ENGS was first noted in 2005. Ideally we would have obtained the same tissues, especially from the gastrointestinal tract, from all cases, in addition to complete medical records and post-mortem examination notes including gross and histological findings. Unfortunately, this was not possible due to the resource-limited setting and resulting opportunistic sampling. Fortunately, the veterinary staff at TCS have begun standardizing sample and record collection for future cases.

Practical and ethical considerations preclude collecting invasive samples from sanctuary chimpanzees, so control samples were limited to serum and feces collected at annual health checks and samples collected post-mortem from individuals determined to have died from causes other that ENGS (e.g., accidental death), which are rare at TCS. Furthermore, the evidence presented here supports an association between "*Ca.* S. troglodytae" and ENGS which, due to ethical considerations, cannot be further investigated by experimentation on chimpanzees. Likewise, infection trials in an animal model (e.g., laboratory mice) would require a pure culture and, as of yet, we are unable to maintain a culture of this bacterium, hence the *Candidatus* designation[52].

We also note that clinically similar cases have not been reported in other captive or wild populations of chimpanzees or other primates. Moreover, despite over 10 years of illness among

the TCS chimpanzees, human cases have not been reported, even among personnel with close daily contact with affected individuals. The genetic and physiological similarities between humans and chimpanzees are often cited as predisposing them to cross-species pathogen exchange[17,111]. It is therefore surprising that no human disease similar to ENGS has been reported to date. Should "*Ca*. S. troglodytae" indeed affect chimpanzees but not humans, it would represent a rare example of such a pathogen[112]. However, we cannot rule out physiological stressors, diet-related factors, environmental conditions, or other pathogens as predisposing factors that differ between humans and chimpanzees. For example, in the case of *C. perfringens*-associated enteritis in humans, changes in gastric and intestinal pH, altered nutritional status, and concurrent infection, particularly with intestinal viruses and parasites, can drastically alter clinical outcomes[113].

To our knowledge, only 44 cases of *Sarcina* infection in humans have been reported in the peer-reviewed literature since the beginning of the 1900s (Supplementary Data 3), and currently no standard treatment for such infections is available. Of published cases treated with at least one specifically mentioned antibiotic (19 of 44), the most common regimen was a combination of oral ciprofloxacin and metronidazole (11 of 19) with a proton-pump inhibitor (8 of 11) or antacid (2 of 11). With the exception of one case involving other complications, treatment was successful when follow-up was noted (9 of 10). Four published cases detail dosages of the antibiotics, all in adult males, and most dosages (3 of 4) were identical: 250 mg metronidazole three times daily and 250 mg ciprofloxacin twice daily for a course of 7 days. Recently, *S. ventriculi* cultured from human blood was shown to be susceptible to other antibiotics including penicillin (minimal inhibitory concentration [MIC] = 0.25 mg/l), amoxicillin (MIC = 0.50 mg/l), amoxicillin–clavulanic acid, piperacillin–tazobactam, imipenem, clindamycin, levofloxacin, rifampicin, vancomycin, and linezolid[104].

Treatment of emphysematous gastritis is similarly unstandardized and includes hemodynamic stabilization with intravenous fluids, broad spectrum intravenous antibiotics effective against Gram-negative and anaerobic bacteria, including meropenem[114,115], cefuroxime and metronidazole[88], nafcillin and cefoxitin[116], and surgery in some cases, but is associated with 60% lethality[103]. That we found evidence of two antibiotic resistance genes in our "*Ca*. S. troglodytae" isolate, a chromosomal *OXA-241*-like gene involved in carbapenem resistance and a plasmid-associated *salA*-like gene linked to lincosamide/streptogramin resistance[67], is noteworthy, as these findings may influence ENGS treatment decisions. Overgrowth of sarcinae in the stomach appears to predispose patients to clinical disease[72]; therefore, probiotics, particularly those containing acidophilic organisms, may prove useful for the treatment or prevention of "*Ca*. S. troglodytae" infections in chimpanzees. For example, probiotics have proven useful for the prevention of *C. difficile*-related disease in humans[117,118]. Finally, autogenous vaccines have proven useful for the prevention of *C. perfringens*-related disease in animals[119,120]. Such an approach could prove useful for the prevention of ENGS if in vitro growth conditions for "*Ca*. S. troglodytae" can be determined.

Since 2011, case studies and reviews concerning *S. ventriculi* and human disease have increased in the medical literature from 0 articles from 1900–2000, to 2 articles from 2000–2010 and 33 articles from 2011—November 2019 (Supplementary Data 3). Increased recent attention to *Sarcina* despite establishment of the genus in 1842 may be coincidental. Alternatively, it may indicate a nascent trend of bacterial emergence[121,122]. The physiological and environmental drivers of *Sarcina* acquisition and subsequent disease progression merit greater attention than they have heretofore received, as does the genetic diversity of the genus. In 34 of 44 published cases, diagnosis of *Sarcina* infection was based on

morphology and/or Gram staining alone with no other diagnostics for confirmation (Supplementary Data 3). Cases of clinical disease associated with *Sarcina* infections should be re-evaluated in light of the possibility that the bacteria identified may represent a complex of cryptic species and strains, some of which are benign but others of which may be highly virulent.

## Methods

**Ethics statement**. TCS located in Western Area National Park, Sierra Leone, is a non-governmental organization that operates under the purview and with the permission of the Ministry of Agriculture, Forestry, and Food Security. All animals originated from Sierra Leone and were confiscated or handed over to TCS under the authority of the Ministry. TCS does not remove any animals from the wild but works to rescue chimpanzees that have been removed from the wild illegally. The care and sampling of resident chimpanzees is officially sanctioned by the Government of Sierra Leone, and samples were shipped to the USA with the official permission of the Government of Sierra Leone under Convention on International Trade in Endangered Species of Wild Fauna and Flora permit number 17US19807C/9.

The presented study was retrospective, did not involve collection of any samples solely for the purpose of this research, and utilized surplus samples collected by TCS veterinarians during routine veterinary procedures and post-mortem examination, which are standard at the sanctuary for any fatality, in compliance with the "Pan African Sanctuary Alliance Primate Veterinary Healthcare Manual"[123] and the policies of TCS.

**Clinical data and samples**. We obtained clinical data from veterinary records for chimpanzees who had died of all causes from 2005 through 2018. These data were compiled by year and by month to make epizootic curves. We then used these data to select samples from the TCS freezer archive according to a case-control study design. Due to resource limitations, samples were collected opportunistically (as opposed to systematically) and archived samples were only available from a subset of cases that occurred from 2013 to 2016. Samples had been collected by staff veterinarians during routine health checks or during post-mortem examination, and samples were fresh-frozen (at −20 or −80 °C) upon collection in whirl packs or test tubes (Supplementary Data 1) and stored long-term at −80 °C. Samples were shipped frozen on dry ice to the United States, stored at −80 °C upon arrival, and kept frozen through processing. To obtain sub-samples of solid tissues and avoid contamination from the external surfaces of organs, we cut frozen tissues with a sterile razor blade and extracted tissue plugs from the newly exposed area with a sterile 6-mm biopsy punch.

**Parasitology**. Microscopy for parasite identification was performed at TCS from 2005 to 2018 following standard veterinary protocols[29]. Briefly, freshly voided fecal samples were collected from individuals and macroscopic features were noted. A direct smear was then made by mixing fecal material with saline and observing the mixture under a light microscope at ×100 and ×400 total magnification, with an additional formalin–ether (10% formalin and ethyl acetate) sedimentation performed as warranted. Slides were read by trained and experienced staff veterinarians. Data on the occurrence of parasites thus identified were compiled from 155 such analyses conducted from 2005 through 2018, representing 17 ENGS-affected chimpanzees (cases) and 13 apparently healthy chimpanzees (controls).

Molecular parasitology using metabarcoding was performed with methods modified from the EMP[32]. DNA was extracted from tissue samples (blood, plasma, serum, lung, and brain) using the DNeasy Blood and Tissue kit (Qiagen, Hilden, Germany) according to manufacturer's instructions and eluted in 50 μl of buffer AE (10 mM Tris–HCl, 0.5 mM ethylenediaminetetraacetic acid). Tissue samples were considered appropriate for this analysis based on published literature showing that infections, including with eukaryotes, can be detected in such samples, even when the tissues analyzed are not the anatomic sites of infection[124,125]. Primers were used to amplify the V9 region of the 18S rDNA gene and were based on published pan-eukaryotic sequences[126,127] (see Supplementary Table 9 for all primers used in this study). These sequences were modified, replacing the individual barcodes with overhang sequences compatible with the Nextera system (Illumina, San Diego, CA, USA)[33–35]. The primers used were EMP_Next_F: 5′-TC GTCGGCAGCGTCAGATGTGTATAAGAGACAGGTACACACCGCCCGTC-3′ and EMP_Next_R: 5′-GTCTCGTGGGCTCGGAGATGTGTATAAGAGACAGT GATCCTTCTGCAGGTTCACCTAC-3′ (IDT, Newark, NJ, USA). To reduce host signal, we used the EMP mammal blocking primer EMP_Mmmal_Block: 5′-GC CCGTCGCTACTACCGATTGGII IIITTAGTGAGGCCCT-[C3 Spacer]-3′ (IDT). PCR reactions were carried out in 25 μl volumes containing 0.3 μM of each primer, 1.6 μM of mammal blocking primer, 12.5 μl 2× HotStart ReadyMix (KAPA Biosystems, Wilmington, MA, USA), and 25 ng template DNA on a C-1000 thermocycler (BioRad, Hercules, CA, USA) with the following cycling conditions: 95 °C for 3 min; 24 cycles of 98 °C for 20 s, 67 °C for 15 s, 62 °C for 30 s, 72 °C for 15 s; and 72 °C for 1 min.

PCR products were purified using the DNA Clean and Concentrator Kit (Zymo Research, Irvine, CA, USA) and eluted in 25 μl of provided elution buffer. From the

25 µl, 5 µl was then used as a template in a 25-µl PCR mix with the Nextera XT Index Kit v2 (Illumina) and limited-cycle PCR for indexing using an annealing temperature of 55 °C with 12 cycles. Products were separated on a 1.5% agarose gel and visualized to confirm band lengths of ~260 bp. Amplicons were then excised from gels and purified using the Zymoclean Gel DNA Recovery Kit (Zymo Research) and eluted in 20 µl of water. Products were quantified using a Qubit fluorometer (Thermo-Fisher Scientific Inc., Waltham, MA, USA). Libraries were sequenced on a MiSeq instrument using paired-end 300 ×300 cycle chemistry (Illumina).

Raw reads were processed with the QIIME v.1.9.1 pipeline[128]. Forward and reverse reads were assembled into paired contigs using the command multiple_join_paired_ends.py and quality filtered using the command multiple_split_libraries_fastq.py with default parameters, except for setting the Phred threshold to 30 or higher (-q 29) and minimum length to 100 bp (-l 100). Chimeras were identified with Usearch v.6.1[129] and removed. Reads were then assigned to OTUs using the QIIME protocol for open reference OTU picking with the command pick_open_reference_otus.py and the default UCLUST tool (v.0.2.0)[129], and taxonomy was assigned to OTUs using default settings with the command assign_taxonomy.py against the SILVA database v. 132[130]. Still-undetermined OTUs were assigned using BLAST within QIIME (-m blast) against the full GenBank database[131] and non-target sequences were then removed by filtering. We processed all data from through all filtering steps after which we removed those samples that represented <0.5% of the total filtered data set from further analyses.

**Virology.** Samples were homogenized by bead beating (for solid tissues), clarified by centrifugation, and treated with nucleases (0.02 U/µl DNAse I, 1 U/µl RNAse T1, 0.04 U/µl RNAse I, 25 ng/µl RNAse A)[36–38]. Viral RNA was isolated using a QIAamp MinElute virus spin kit (Qiagen), omitting carrier RNA. Extracted nucleic acids were then converted to double-stranded cDNA using the SuperScript double-stranded cDNA Synthesis Kit (Invitrogen, Carlsbad, CA, USA) and random hexamers and purified using Ampure XP beads (Beckman Coulter, Brea, CA, USA). Approximately 1 ng of DNA was prepared as a library for pair-ended sequencing on a MiSeq instrument (MiSeq Reagent kit v3, 150 cycle) using the Nextera XT DNA Library Prep Kit (Illumina). Sequence data were analyzed using CLC Genomics Workbench version 11.0 (Qiagen). In brief, we trimmed low-quality bases (Phred quality score <30), discarded short reads (<75 bp), and subjected the remaining reads to de novo assembly using the CLC assembler with automatic word and bubble size selection and a minimum contig length of 500. We then analyzed contigs for nucleotide-level (blastn) and protein-level (blastx) similarity to known viruses in GenBank. For blastx we applied the BLASTX algorithm with the BLOSUM62 matrix to sequences translated into all six frames. We also analyzed all sequence data at the individual read level by mapping reads to viruses in the GenBank database using the CLC mapping tool at low stringency (length fraction of 0.5, similarity fraction of 0.6).

**Bacterial metabarcoding.** Genomic DNA was extracted from solid tissue and blood samples using a DNeasy Blood and Tissue Kit and from fecal, vomit, and stomach content samples using a DNeasy PowerSoil DNA Isolation Kit (Qiagen) according to manufacturer's instructions. The V4 region of the bacterial 16S rRNA gene was amplified using the following primers, which contain the Illumina sequencing adapters, an 8-nt indexed barcode sequence, a 10-nt pad sequence, a 2-nt linker, and the 16S rRNA V4-specific primer: 16SV4_F: 5′-AATGATACGGC GACCACCGAGATCTACACNNNNNNNNNTATGGTAATTGTGTGCCAGCMG CCGCGGTAA-3′ and 16S_V4_R: 5′-GGACTACHVGGGTWTCTAATCCAGT CAGTCAGNNNNNNNNNCAAGCAGAAGACGGCATACGAGAT-3′[132]. PCR reactions were carried out in 25 µl volumes containing 0.3 µM of each primer, 12.5 µl 2× HotStart ReadyMix (KAPA Biosystems), 6.5 µl water, and 25 ng template DNA with the following cycling conditions: 95 °C for 3 min; 30 cycles of 95 °C for 30 s, 55 °C for 30 s, 72 °C for 30 s; and 72 °C for 5 min. PCR products were then electrophoresed on 1% low melt agarose gels (National Diagnostics, Atlanta, GA, USA), excised, purified using a ZR-96 Zymoclean Gel DNA Recovery Kit (Zymo Research), and quantified using a Qubit® Fluorometer (Thermo Fisher Scientific). Equimolar amounts of the barcoded V4 amplicons were pooled and sequenced using a MiSeq 2 × 250 bp v2 kit (Illumina) using custom sequencing primers with 10% PhiX control DNA.

All sequences were demultiplexed on the Illumina MiSeq and were processed and analyzed using mothur v.1.42.0.[132]. Poor quality sequences were removed after paired end sequences were combined into contigs. Sequences were aligned against the SILVA 16S rRNA gene reference alignment database to screen for alignment to the correct region. Preclustering was performed to reduce error and chimeras were detected and removed using UCHIME v.4.2[133]. The SILVA database was used to classify bacterial sequences while sequences classifying to mitochondria, cyanobacteria, Eukarya, Archaea, or Fungi were removed along with singletons to streamline analysis. Percent abundance graphs were created in GraphPad Prism v8.4.3 (GraphPad Software, San Diego, CA, USA).

**Diagnostic PCR.** PCR primers were designed to the V2–V3 region of the "Ca. S. troglodytae" 16S rRNA gene: TacuSarc_Diag_F: 5′-TGAAAGGCATCTTTTAAC AATCAAAG-3′ ($T_m$ = 52.8 °C) and TacuSarc_Diag_R: 5′-TACCGTCATTATCG TCCCTAAA-3′ ($T_m$ = 53 °C) (IDT). PCR reactions were carried out in 25 µl

volumes containing 0.2 µM of each primer, 12.5 µl 2× HotStar Master Mix (Qiagen), 10 µl water, and 25 ng template DNA on a C-1000 thermocycler (BioRad, Hercules, CA, USA) with the following cycling conditions: 95 °C for 15 min; 29 cycles of 94 °C for 30 s, 48 °C for 30 s, 72 °C for 30 s; and 72 °C for 10 min. PCR products (289 bp expected length) were then electrophoresed on 1.5% low-melt agarose gels with ethidium bromide and 1 kb plus DNA length standards (BioRad), visualized under UV light, and photographed using a GelDoc XR imager (BioRad). Amplicons were then excised and purified as described above and Sanger sequenced on ABI 3730xl DNA Analyzers (Applied Biosystems, Foster City, CA, USA) at the University of Wisconsin-Madison Biotechnology Center.

**Bacterial isolation and characterization.** Liquid samples were pipetted directly onto sterile agar plates and solid tissues (deep cut sections collected with a 6-mm biopsy punch to avoid external contamination) were placed in sterile Petri dishes and minced with two sterile blades until homogenized. 200 µl of pre-reduced thioglycollate medium (Hardy Diagnostics, Santa Maria, CA, USA) was added and the mixture was streaked by inoculating loop onto a 100 mm × 15 mm plate, placed immediately into an Anaerogen Compact anaerobic pouch (Oxoid Limited, Hampshire, UK), sealed, and incubated at 37 °C. For liquid growth media, a sterile 18-gauge needle with 1-ml syringe was used to inoculate stoppered tubes that were incubated anaerobically at 37 °C. Cultures were screened by PCR, and cells were directly visualized and grown at least 10 days before they were deemed negative for growth of the bacterium of interest.

For comparison, the type strain *S. ventriculi* "Goodsir" (ATCC 19633 or ATCC 29068) was obtained from the ATCC (American Type Culture Collection, Manassas, VA, USA) and grown according to ATCC guidelines.

**Bacterial imaging.** Live bacterial cells diluted in sterile water or phosphate buffered saline were placed on glass slides, examined with light microscopy, and imaged immediately. For Gram staining, heat-fixed slides were flooded with crystal violet solution for 1 min, rinsed with water, flooded with iodine solution for 1 min, rinsed with water, flooded with decolorizer solution for 1–5 s, rinsed with water, counterstained with five drops of safranin solution for 30 s, rinsed with water, and air dried. For methylene blue staining, heat-fixed slides were flooded with 1% aqueous solution of methylene blue for 1 min at room temperature, then washed with distilled water and then air dried. All slides were visualized and photographed at ×400 on a Panthera U microscope with a Moticam 5.0 camera (Motic, British Columbia, Canada). For cell size measurements, strains JB1 and ATCC 29068 "Goodsir" were plated and grown on SVGM plates under identical conditions for 72 h. Bacterial cells from seven distinct colonies were harvested, blinded to the investigator, and examined as follows: live cells in phosphate buffered saline from five non-overlapping visual fields were captured as above and single-cell diameters were quantified using the circle (3-point) measurement tool in the Images Plus software suite v.2.0 (Motic). Cellular diameters were compared using a Mann–Whitney $U$ test (two-tailed).

Tissues for histopathology were collected during post-mortem examination by staff veterinarians and immediately fixed in 4% paraformaldehyde at least overnight, then later dehydrated in alcohol, embedded in paraffin wax, cut in 6-µm sections, stained with hematoxylin and eosin, visualized under a light microscope, and photographed. For direct visualization of brain tissue, a 6-mm biopsy punch (Integra LifeSciences, Plainsboro, NJ, USA) was taken from the interior of the cerebrum, minced with sterile blades, smeared onto a clean glass slide, and immediately imaged as described above.

**16S rDNA phylogeny.** The full 16S rDNA sequence from "*Ca.* S. troglodytae" isolate JB1 (1508 bp) was queried against the NCBI 16S ribosomal RNA sequence (*Bacteria* and *Archaea*) database using megablast[131] with default parameters, and the top 50 results as of 17 July 2019 were downloaded from NCBI's RefSeq or GenBank (all *e*-values 0). For taxonomic completeness, 73 sequences comprising "cluster I" *Clostridia*[134] as previously published[135] were also retrieved from GenBank, and duplicates were removed. The type organism from the closest known clade (*Hathewaya histolytica*, located in "cluster II" of the clostridia as defined by Collins et al. 1994[134]) was included as an outgroup[135]. The resulting 98 sequences, plus the "*Ca.* S. troglodytae" sequence, were aligned using MUSCLE3.8.31[136] (final alignment length 1585 positions). To quantify nucleotide-level distances among sequences, a pairwise distance matrix was calculated using MEGA7 v.7.0.96[137] with pairwise deletion and 1000 bootstrap replicates to estimate standard errors. The phylogenetic position of "*Ca.* S. troglodytae" was then inferred with PhyML v. 1.8.1[138] using the general time reversible (GTR) substitution model as determined by Smart Model Selection[139], and 1000 bootstrapped data sets were used to estimate statistical confidences of clades.

**Whole-genome sequencing, assembly, and annotation.** A large, single colony of cells morphologically consistent with "*Ca.* S. troglodytae" (Fig. 5a) which tested positive by diagnostic PCR was grown for 7 days on an SVGM plate at 37 °C in an anaerobic pouch. The entire colony ("isolate JB2") was transferred into a sterile 1.5-ml tube and genomic DNA was extracted using the Wizard Genomic DNA purification kit (Promega) according to the manufacturer's instructions. Mate-pair sequencing libraries were constructed using 1 µg of resulting DNA and the Nextera

gel-free protocol and quantified using NEBNext qPCR (New England Biolabs, Ipswich, MA, USA). All libraries were loaded in equimolar amounts, multiplexed, and sequenced on an Illumina MiSeq using the $2 \times 300$ bp V3 chemistry.

The mate-pair reads were processed with Nxtrim v.0.4.4[140], then processed with bbduk v.38.72[141] using Phred = 20, length >50 base pairs, and subsampled to 100× coverage with bbnorm v.3.8.9[142]. We used bowtie2 v.2.3.5[143] to remove sample-to-sample bleed through. All reads were assembled with SPAdes v.3.10.1[53] using -careful and -mp arguments and manually closed with Bandage v.0.8.1[144], CLC Workbench 12.0 and EDGE Bioinformatics v.2.3.1[145]. The PATRIC server v.3.6.2[146] with default settings was used for all ORF annotations, CLC Genomics Workbench 12.0 was used for variant analysis, Mauve v.2.4.0[147] was used for genome comparisons, ISFinder[148] was used for insertion sequence identification, the Resistance Gene Identifier (RGI v.4.2.2) from the Comprehensive Antibiotic Resistance Database (CARD v.3.0.0)[65] was used for antibiotic resistance gene identification, a customized database from the Virulence Factor Database (VFD)[64] was used for virulence factor identification with ShortBRED v.0.9.5[63], and average nucleotide identity (ANI) calculator (Enveomics v.0.1.3)[55] was used to generate the average nucleotide identities.

**Reporting summary**. Further information on research design is available in the Nature Research Reporting Summary linked to this article.

## Data availability

Sequence data that support the findings of this study have been deposited in the National Center for Biotechnology Information (NCBI) GenBank database with the accession codes CP051754, CP051755, CP051756, CP051757, CP051758, CP051759, CP051760, CP051761, CP051762, CP015763, CP015764 (Bacterial genome assembly) and MT350347, MT350348, MT350349, MT350350, MT350351, MT350352, MT350353, MT350354, MT350355, MT350356, MT350357 (viral replicase genes) and in the NCBI Sequence Read Archive with BioProject accession codes PRJNA648419 (16S and 18S metabarcoding reads) and PRJNA625238 (Bacterial genome sequencing reads). Source data are provided with this paper.

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

## Acknowledgements

We gratefully acknowledge and thank Tacugama Chimpanzee Sanctuary founder Bala Amarasekaran and veterinarian Dr. Rosa Garriga for their efforts in identifying and investigating ENGS, and the entire TCS staff for collection, storage, and coordination of shipping of samples, and for their dedication to the health and conservation of chimpanzees. We also gratefully acknowledge Heather N'te Inzalaco, Travis Wentz, Sabine Pellett, and Eric Johnson at UW-Madison (WI, USA) for providing bacterial growth media and staining reagents and for advice regarding bacterial culture. We also thank Blanca Pérez and Dr. Alberto Marco Valle at UD Histologia i Anatomia Patològica, Facultat de Veterinària, Universitat Autònoma de Barcelona, Bellaterra (Barcelona, Spain) for Gram staining of selected ENGS-affected chimpanzee paraffin-embedded tissues. We are grateful to Thomas Postler at Columbia University (NY, USA) for his knowledge of classic linguistics and guidance in naming the novel bacterium. We thank Laura Bollinger (IRF-Frederick) for editing the manuscript. The content of this publication does not necessarily reflect the views or policies of the U.S. Department of Health and Human Services, Department of the Navy, Department of Defense, or the institutions and companies affiliated with the authors. K.A.B.-L. is an employee of the U.S. Government and L.A.E. is a military service member. This work was prepared as part of their official duties. Title 17 U.S.C. §105 provides that 'Copyright protection under this title is not available for any work of the United States Government.' Title 17 U.S.C. §101 defines a U.S. Government work as a work prepared by a military service member or employee of the U.S. Government as part of that person's official duties. This work was supported in part through the prime contract of Laulima Government Solutions, LLC, with the U.S. National Institute of Allergy and Infectious Diseases (NIAID) under contract no. HHSN272201800013C and Battelle Memorial Institute's former prime

contract with NIAID under contract no. HHSN272200700016I. J.H.K. performed this work as a former employee of Battelle Memorial Institute and a current employee of Tunnell Government Services (TGS), a subcontractor of Laulima Government Solutions, LLC, under contract no. HHSN272201800013C. This research was also supported by Global Emerging Infections Surveillance (GEIS) project P0071_19_NM_06 (K.A.B.-L.) and WUN A1417, the University of Wisconsin-Madison Parasitology and Vector Biology Training Program (to L.A.O.) and the University of Wisconsin-Madison John D. MacArthur Professorship Chair (to T.L.G.).

## Author contributions

T.L.G. conceived the study; L.A.O., T.L.G., and I.H. designed the study; I.H., A.P., J.E.J., and S.M. coordinated sample collection and transport; L.A.O., B.C., K.A.B.-L., L.J.V., C.L.D., C.D.D., and C.J.-S. performed laboratory work and collected data; L.A.O., K.A.B.-L., L.A.E., L.J.V., J.H.K., G.S., C.L.D., analyzed data; L.A.O. and T.L.G. wrote the manuscript; all authors made substantive intellectual contributions, revised the manuscript, and approved the final draft.

## Competing interests

The authors declare no competing interests.
