## [Peer Review File · Nature Communications]

REVIEWER COMMENTS

Reviewer #1 (Remarks to the Author):

The paper from Owens and colleagues describes a novel bacterium in the *Sarcina* genus that is associated with mortality in chimpanzees in a sanctuary in Sierra Leona. The chimpanzees that died had a syndrome named "epidemic neurologic and gastroenteric syndrome" abbreviated as ENGS. The paper is of interest and the laboratory work conducted to identify a pathogen associated represents a lot of work.

Since 2005, 56 chimpanzees died from this disease, and apparently higher rates of deaths are observed in certain periods of the year. It would be of interest for the reader to have more information on this disease, in addition to the clinical signs, for example, how long are animals sick before they die, is it going gradually, etc., is it occurring in several individuals at the same time, in animals living in close proximity etc..

What are the results for other parameters, haematology, biochemistry in these animals. How long were the animals in the sanctuary before they developed disease?

Although the laboratory findings are interesting and described in detail, it is not at all clear to the reader in which organs and/or fluids the bacteria was identified, and which samples were analyzed for what. Where all samples analyzed for viruses and bacteria? On which kind of samples were parasitological tests performed. There is thus a need to have a table that shows what has been done for each animal with the corresponding results for the new bacteria. In which animals and which organs or fluids the virus was subsequently amplified by PCR.

To what extent can the authors be sure that this bacteria is the cause or if the presence is just associated with a potential other course.

Did the authors compare virome, microbiome of sick versus healthy animals? These data should also be presented if available

The paper needs to be improved by this additional information, and in particular, the authors should show clearly details on all organs and fluids animals per animal versus controls; which could strengthen their claim on potential pathogenicity and cause of the bacteria in ENGS in the chimpanzees from the sanctuary.

Are the authors aware of similar observations (ENGS) in other sanctuaries in Africa that house chimpanzees, but maybe also other ape species like gorillas or bonobos? o

Reviewer #2 (Remarks to the Author):

In their manuscript entitled "A novel bacterium in the genus *Sarcina* linked to a uniformly lethal epizootic disease in sanctuary chimpanzees (*Pan troglodytes verus*) in Sierra Leone", Owens and colleagues describe the identification of a putatively new species of the genus *Sarcina*. The bacterium tentatively named "*Candidatus Sarcina troglodytae*" was discovered when searching for the causative agent of a syndrome the authors named "epidemic neurologic and gastroenteric syndrome (ENGS)" which caused the deaths of at least 56 chimpanzees. The authors analysed a substantial number of samples in order to identify the pathogen and rule out other potential pathogens as causes of death. These analyses also comprised parasite metabarcoding and metagenomics for the detection of viruses. From the results of these analyses the authors rule out pathogens other than the described "*Ca. S. troglodytae*" being the cause of death. Although overall the presented results are convincing, the conclusions of the study should be strengthened by improving the parts dealing with sequencing-based analyses since on that basis the authors rule out other pathogens. This strengthening would also be important, because the authors discuss (lines 429 – 431) that "For example, in the case of *C. perfringens*-associated enteritis in humans, changes in gastric and intestinal pH, altered nutritional status, and concurrent infection,

particularly with intestinal viruses and parasites, can drastically alter clinical outcomes." Therefore, I propose at least the following for the sequencing-based analyses:

1. The authors should add detailed information in the supplement about the results obtained from parasite metabarcoding. I am missing results on the total raw dataset sizes obtained, the effect of the read processing steps, especially quality filtering (numbers of trimmed bases, filtered reads) and distribution of the trimmed reads between host and parasites. This is an important information to assess the meaningfulness of the obtained data. This is especially important since the cited proven protocol was modified for compatibility with the Illumina Nextera system. This modification changed the overhang of the primers (lines 521-522). Changes in primer overhangs have been shown to impact primer binding and eventually bias the PCR.

In addition, why was the threshold of a minimum of 150 reads per sample (line 549 "any samples with fewer than 150 reads were removed") set and how is this number justified? Was it really applied to samples or rather to OTUs (there was reportedly no sample with less than 681 reads)? In case the threshold was indeed for the OTU, with the smallest datasets that remained after filtering, 150 reads would make up roughly 20% of the remaining reads. That doesn't appear well justified to be deleted. On the other hand, in case it is really aimed at samples, then this should also take into account the overall dataset and from my point of view should be a proportion of the total data.

2. Virology: Supplementary Table 3 is incomplete. Maybe in the part that is hidden there is the relevant information ... please correct that.

As it stands, a lot of information for the metagenomics for virus detection is missing. As for the parasite metabarcoding, information on datasets (raw reads, reads after trimming/filtering, number and sizes of contigs, sequence depths of the contigs, numbers of unassembled reads) are missing, again making it hard to assess the results. For instance, the size of a dataset strongly impacts the sensitivity of the analysis, therefore this is very important information.

Please specify the parameters used for sequence assembly and BLAST analyses.

Moreover, there are meanwhile numerous studies where only few reads were sufficient as a first hint for detection of the respective causative virus. Hence, the datasets should be analysed at the read level to ensure there is no virus hidden, that could be a cofactor of the disease (see above). It has also been shown in a number of studies that the selection of the sample strongly impacts the chance to detect a virus. Despite this knowledge, in the present paper, information regarding the samples selected for metagenomics sequencing for the search of viruses is missing.

The abovementioned information needs to be added in order to show that the author's notion that "no viruses were case-associated" (lines 155/156) and "we were able to rule out eukaryotic parasites and viruses as causes of ENGS." (lines 320/321) is valid.

3. Bacterial metabarcoding: As for parasite metabarcoding and virus metagenomics, please report details of the obtained datasets and the results of pre-processing the data. As for the other sequencing analyses, please add information which samples were analysed. Maybe this can all be added in supplementary table 1 by highlighting the selected tissues or adding an extra column with samples used for each of the sequencing analyses.

In lines 173-175 the authors state "Re-examination of all samples by including the *Clostridium* (*Sarcina*) *ventriculi* sequence from GenBank in the Greengenes database demonstrated this organism to be present in all 22 ENGS case samples." Please provide data on the abundance of *Clostridium* (*Sarcina*) *ventriculi*, or is this meant to be shown in Fig 3b? There, *Sarcina* is invisible in 3 of the 22 bars (Bebi Blood, Bbebi Heart, Bebi Muscle).

Further comments:

4. It would be helpful for the reader to have a graphic display of the groups of cases and controls and symptoms for the data presented in section "Epizootiology, clinical signs, and pathology" (compare for instance Figure 1 in [https://doi.org/10.1016/S1473-3099\(19\)30546-8](https://doi.org/10.1016/S1473-3099(19)30546-8)). The data presented in section "Samples" could also be included in a graphic since the composition of the sample panel and the relations of the samples with the individuals is not entirely clear from what is presented in the text.

5. The statement "The chimpanzees in this study [...] were sampled between 14 March 2013 and 11 July 2016." is contradictory with the presentation in Figure 1 where samples from before 2013 are included.

6. Numbers of examinations and group sizes (cases/controls) in the results presented on parasitology differ from the methods:

Results/Parasitology (lines 134-136): Microscopic examinations of fecal samples were performed

on site for 30 chimpanzees (17 ENGS cases and 13 controls) from 2005 - 2018 using standard direct and sedimentation methods, comprising 155 analyses with a median 5 analyses per chimpanzee.

Methods/Parasitology (lines 509-512): Data on the occurrence of parasites thus identified were compiled from 117 such examinations conducted from 2005 through 2018, representing 13 ENGS-affected chimpanzees (cases) and 17 apparently healthy chimpanzees (controls).

Minor comments:

- Suppl. Tab 1: It would greatly enhance the readability if information in column "Signs" was stated in clear text instead of the number code. The same is true for the abbreviations for the tissues. Maybe the authors could use landscape format for this table to gain enough space for this information. Please also consider including information on the samples used for the different analyses (see above).
- Suppl. Tab. 2: "Fungi" is no genus; numbers in brackets are no percentages as stated in footnote a
- Spelling of "Ole" in Suppl. Fig 2 and Suppl. Tab. 1 differs ("Ole" vs. "Olé")
- Please check for "gastric dilation" and "gastric dilatation" (for example, in the sentence in lines 370 - 373 both appear to be used synonymously).
- In line 443 a closing ")" is missing after "MIC = 0.50 mg/l"

Reviewer #3 (Remarks to the Author):

NCOMMS-20-24756 A novel bacterium in the genus *Sarcina* linked to a uniformly lethal epizootic disease in sanctuary chimpanzees (*Pan troglodytes verus*) in Sierra Leone

The manuscript describes the identification of a new bacterial species associated with acute to peracute death in chimpanzees. The manuscript will likely be of interest to the readership.

The main challenges in this manuscript relate to the definition of cases, lack of data points for many cases, and the inconsistent sampling. Much of this may be due to the sanctuary setting and problems associated with opportunistic sampling.

Case definition: The 56 cases that comprise "epidemic neurologic and gastroenteric syndrome" (ENGS) appear to be of two distinct types. About 60% (32 cases) have varied neurologic and gastric symptoms; the other 40% (24 cases) appear to be sudden death or found dead animals, without prior signs. Although several clostridial organisms can do this, initially combining these into one syndrome (before the bacterial findings) seems a bit tenuous.

Was ENGS initially identified as both symptomatic and sudden death animals, or did this realization occur over time?

Post mortem evaluation: Post mortem evaluations were only available for 19 animals, with only 14 of these having the gastrointestinal tract examined, and only 11 with gastric distention. It would seem that this would represent a much better-defined group to initially focus on for the various assays. After finding the bacteria, then the thought would occur to evaluate the sudden death cases. The authors could then introduce the findings in the sudden death cases to show the bacteria was also present in these cases and suggest that they are variations in presentation. Logically, that is how I would have assumed this process to have occurred.

Samples: Based on Supplementary Table 1, actual tissue samples were only obtained from 7 cases and 1 control. All other samples were blood or body fluids. It would be best to use GI tissues, assuming they are available.

Samples II: It is hard to assimilate which of the various assays were performed for the various samples. Were all samples evaluated for all assays (Parasitology, virology, etc?). I am pretty certain they were not, however a table listing all the samples and which ones were tested by each assay (and which were positive) would greatly improve interpretation by the reader.

In the virology section, it states "no viruses were case-associated". I am assuming that means that no viruses were found in "syndrome" cases, just in controls? It was unclear, at least to me.

I strongly agree with the authors (lines 373-376' that "A more consistent gross and histopathologic evaluation of affected chimpanzees in the future" is needed to better define the syndrome. However, I think that can be addressed in later studies.

Although the above may sound negative, overall, I feel the authors have done a good job in trying to present a hard to define syndrome using the materials and data available. My major issues involve addressing more clearly the areas above to more accurately describe the various caveats and gaps in the data. I find their overall premise and reasoning (and the data they have) consistent with their claims.

“A novel bacterium in the genus *Sarcina* linked to a uniformly lethal epizootic disease in sanctuary chimpanzees (*Pan troglodytes verus*) in Sierra Leone”

For publication in *Nature Communications*

Point-by-point responses to reviewer comments

Reviewer #1 (Remarks to the Author):

Comment 1: *The paper from Owens and colleagues describes a novel bacterium in the Sarcina genus that is associated with mortality in chimpanzees in a sanctuary in Sierra Leone. The chimpanzees that died had a syndrome named “epidemic neurologic and gastroenteric syndrome” abbreviated as ENGS. The paper is of interest and the laboratory work conducted to identify a pathogen associated represents a lot of work.*

Response 1: Many thanks for the positive comments and for recognizing the effort involved in this study.

Comment 2: *Since 2005, 56 chimpanzees died from this disease, and apparently higher rates of deaths are observed in certain periods of the year. It would be of interest for the reader to have more information on this disease, in addition to the clinical signs, for example, how long are animals sick before they die, is it going gradually, etc., is it occurring in several individuals at the same time, in animals living in close proximity etc..*

What are the results for other parameters, haematology, biochemistry in these animals. How long were the animals in the sanctuary before they developed disease?

Response 2: We are grateful to the reviewer for making this suggestion. To address this point, we have examined additional clinical records from ENGS cases and have summarized the requested information in the form of a new table entitled “Disease Characteristics” (Supplementary Table 1) and have added corresponding text in the Results Section (lines 106-108, 117-119). We note that certain parameters are missing for certain animals, due to the resource-limited setting in which cases occurred. Fortunately, digital record-keeping and limited diagnostics are now in use at the sanctuary, so we plan to collect these data for future cases and follow-up studies. We have added text to this effect to the Discussion (lines 462-463).

Comment 3: *Although the laboratory findings are interesting and described in detail, it is not at all clear to the reader in which organs and/or fluids the bacteria was identified, and which samples were analyzed for what. Where all samples analyzed for viruses and bacteria? On which kind of samples were parasitological tests performed. There is thus a need to have a table that shows what has been done for each animal with the corresponding results for the new bacteria. In which animals and which organs or fluids the virus was subsequently amplified by PCR.*

Response 3: This is indeed a valid point. Based on the reviewer's suggestion, we have compiled the full information for each sample and added a new table entitled "Comprehensive Testing Results" to the Supplemental Information (Supplementary Table 3).

Comment 4: *To what extent can the authors be sure that this bacteria is the cause or if the presence is just associated with a potential other course.*

Response 4: We agree that this is an important question. In writing the manuscript, we aimed to be judicious in our word choice to emphasize that our data support an association, with further studies required to assess causation, and that we cannot unequivocally rule out another cause. We have further toned down the wording in the Discussion so as to be cautious about this issue and have therefore changed "we were able to rule out eukaryotic parasites and viruses as causes of ENGS" to "we found no differences between case and control groups with regards to infection with any parasite or virus and therefore deemed these types of organisms unlikely to be causes of ENGS" (lines 366-368) and changed "The precise role of '*Ca. S. troglodytae*' in the etiology of ENGS..." to "The potential role of '*Ca. S. troglodytae*' in the etiology of ENGS..." (line 455-456).

Comment 5: *Did the authors compare virome, microbiome of sick versus healthy animals? These data should also be presented if available*

Response 5: We thank the reviewer for raising this important point. We have indeed performed these comparisons. For clarification, we have moved the statistical analyses out of the Supplementary Tables (removing part of Supplementary Table 3 and all of Supplementary Table 2), added statistical analysis for the diagnostic "*Ca. S. troglodytae*" PCR, and combined these data to create a new summary table of results for cases versus controls, entitled "Case/Control Statistical Summary" (Supplementary Table 5).

Comment 6: *The paper needs to be improved by this additional information, and in particular, the authors should show clearly details on all organs and fluids animals per animal versus controls; which could strengthen their claim on potential pathogenicity and cause of the bacteria in ENGS in the chimpanzees from the sanctuary.*

Response 6: We agree. As described above, we have created two additional tables with the details suggested by the reviewer: Supplementary Table 3 ("Comprehensive Testing Results"), which displays each test result for every sample (including all cases and controls), and Supplementary Table 5 ("Case/Control Statistical Summary"), which displays case versus control results and corresponding *P*-values for each assay.

Comment 7: *Are the authors aware of similar observations (ENGS) in other sanctuaries in Africa that house chimpanzees, but maybe also other ape species like gorillas or bonobos? o*

Response 7: Great question! We have performed a comprehensive literature review and have not discovered similar diseases in sanctuary or wild populations of great apes of any

species. We have also inquired with colleagues involved in primate veterinary medicine and chimpanzee sanctuary work and none of them have never encountered a disease like ENGS. We have added a statement to the Discussion to this effect (lines 473-474).

Reviewer #2 (Remarks to the Author):

Overall comment: *the conclusions of the study should be strengthened by improving the parts dealing with sequencing-based analyses since on that basis the authors rule out other pathogens. This strengthening would also be important, because the authors discuss (lines 429 – 431) that “For example, in the case of C. perfringens-associated enteritis in humans, changes in gastric and intestinal pH, altered nutritional status, and concurrent infection, particularly with intestinal viruses and parasites, can drastically alter clinical outcomes.” Therefore, I propose at least the following for the sequencing-based analyses.*

Response: The reviewer makes an excellent point. We address this critique in the comments that follow.

Comment 1: *The authors should add detailed information in the supplement about the results obtained from parasite metabarcoding. I am missing results on the total raw dataset sizes obtained, the effect of the read processing steps, especially quality filtering (numbers of trimmed bases, filtered reads) and distribution of the trimmed reads between host and parasites. This is an important information to assess the meaningfulness of the obtained data. This is especially important since the cited proven protocol was modified for compatibility with the Illumina Nextera system. This modification changed the overhang of the primers (lines 521-522). Changes in primer overhangs have been shown to impact primer binding and eventually bias the PCR. In addition, why was the threshold of a minimum of 150 reads per sample (line 549 “any samples with fewer than 150 reads were removed”) set and how is this number justified? Was it really applied to samples or rather to OTUs (there was reportedly no sample with less than 681 reads)? In case the threshold was indeed for the OTU, with the smallest datasets that remained after filtering, 150 reads would make up roughly 20% of the remaining reads. That doesn't appear well justified to be deleted. On the other hand, in case it is really aimed at samples, then this should also take into account the overall dataset and from my point of view should be a proportion of the total data.*

Response: We thank the reviewer for the careful consideration of our parasite metabarcoding methods. Based on the reviewer's suggestions, we have added the total number of reads generated and % of reads filtered to the Results, Parasitology section (lines 159-163) and have added the additional requested sequencing statistics to a new table (Supplementary Table 4). In addition, we have added text to the Results, Parasitology section that directly addresses the issue of host reads in our data (lines 163-166). We have also clarified in the same section that the primers used (though modified from the original publication which reported the sequence-specific portion) have previously been used in this modified form in several other studies (line 160). In regard to the minimum threshold: this was indeed reported in an unclear way, and we thank the reviewer for pointing out this omission. The threshold was set for the samples (not

OTUs) and was, as the reviewer suggested, based on the proportion of the total data set. To address this point, we have added the details requested by the reviewer to the Results, Parasitology section (lines 166-170) and the Methods, Parasitology section (lines 604-605).

Comment 2: *Virology: Supplementary Table 3 is incomplete. Maybe in the part that is hidden there is the relevant information ... please correct that. As it stands, a lot of information for the metagenomics for virus detection is missing. As for the parasite metabarcoding, information on datasets (raw reads, reads after trimming/filtering, number and sizes of contigs, sequence depths of the contigs, numbers of unassembled reads) are missing, again making it hard to assess the results. For instance, the size of a dataset strongly impacts the sensitivity of the analysis, therefore this is very important information. Please specify the parameters used for sequence assembly and BLAST analyses. Moreover, there are meanwhile numerous studies where only few reads were sufficient as a first hint for detection of the respective causative virus. Hence, the datasets should be analysed at the read level to ensure there is no virus hidden, that could be a cofactor of the disease (see above). It has also been shown in a number of studies that the selection of the sample strongly impacts the chance to detect a virus. Despite this knowledge, in the present paper, information regarding the samples selected for metagenomics sequencing for the search of viruses is missing. The abovementioned information needs to be added in order to show that the author's notion that "no viruses were case-associated" (lines 155/156) and "we were able to rule out eukaryotic parasites and viruses as causes of ENGS." (lines 320/321) is valid.*

Response: We agree that these are important points. To address the issue of missing information, we have modified the virology results table such that all columns are now visible (Supplementary Table 6) and have added the sequencing statistics as the reviewer requests to the Results, Virology section (lines 176-185). We also added assembly and BLAST parameters to the Methods, Virology section (lines 618-619 and 621-622) and specified exactly which samples were used for analyses as part of a newly-created table (Supplementary Table 3; see response to Reviewer 1, above). The reviewer makes an intriguing point regarding analysis at the read level. We have therefore carefully reanalyzed the data according to the method specified by the reviewer and found no evidence of additional viruses. We have added the details of this analysis to the Methods, Virology section (lines 622-624) and the outcome to the Results, Virology section (lines 185-186).

Comment 3: *Bacterial metabarcoding: As for parasite metabarcoding and virus metagenomics, please report details of the obtained datasets and the results of pre-processing the data. As for the other sequencing analyses, please add information which samples were analysed. Maybe this can all be added in supplementary table 1 by highlighting the selected tissues or adding an extra column with samples used for each of the sequencing analyses. In lines 173-175 the authors state "Re-examination of all samples by including the Clostridium (Sarcina) ventriculi sequence from GenBank in the Greengenes database demonstrated this organism to be present in all 22 ENGS case samples." Please provide data on the abundance of Clostridium (Sarcina) ventriculi, or is this meant to be shown in Fig 3b? There, Sarcina is invisible in 3 of the 22 bars (Bebi Blood, Bbebi Heart, Bebi Muscle).*

Response: We are grateful to the reviewer for making these suggestions. To address the issue of missing information, we have added the requested sequencing statistics to the Results, Bacterial Metabarcoding section (lines 197-202). As requested, we have added the missing values for percent abundance of *Sarcina* as numbers over each bar in panel b of Figure 4. We also put these percent abundance values, along with specification of exactly which samples were used for analyses, into a newly-created table (Supplementary Table 3; see response to Reviewer 1, above).

Further comments:

Comment 4: *It would be helpful for the reader to have a graphic display of the groups of cases and controls and symptoms for the data presented in section “Epizootiology, clinical signs, and pathology” (compare for instance Figure 1 in [https://doi.org/10.1016/S1473-3099\(19\)30546-8](https://doi.org/10.1016/S1473-3099(19)30546-8)). The data presented in section “Samples” could also be included in a graphic since the composition of the sample panel and the relations of the samples with the individuals is not entirely clear from what is presented in the text.*

Response: We had not considered this possibility; this is an excellent suggestion. We compiled clinical data for all cases (data presented in the Epizootiology, Clinical Signs, and Pathology section), compiled clinical data for the subset of cases used in the study (data presented in the Samples section), and have created a graphic as the reviewer suggests. We have added the suggested image to the manuscript as the new Figure 1 to orient the audience and serve as a reference point while reading through the paper.

Comment 5: *The statement “The chimpanzees in this study [...] were sampled between 14 March 2013 and 11 July 2016.” is contradictory with the presentation in Figure 1 where samples from before 2013 are included.*

Response: Great point; this was indeed not clear as presented. To address this issue, we now clearly show in Figure 1 that samples are a subset of total cases by placing a bar under the x-axis to represent the period of time from which samples were collected (Figure 2). Furthermore, we have added text to the Results, Samples section indicating that the samples are a subset of the total cases (line 134) and that not all cases in that time period were sampled (line 141). We state in the Methods that samples were collected opportunistically (not systematically) beginning in 2013 and not every case was sampled due to lack of resources (lines 548-550).

Comment 6: *Numbers of examinations and group sizes (cases/controls) in the results presented on parasitology differ from the methods:*

Results/Parasitology (lines 134-136): Microscopic examinations of fecal samples were performed on site for 30 chimpanzees (17 ENGS cases and 13 controls) from 2005 - 2018 using standard direct and sedimentation methods, comprising 155 analyses with a median 5 analyses per chimpanzee.

Methods/Parasitology (lines 509-512): Data on the occurrence of parasites thus identified were compiled from 117 such examinations conducted from 2005 through 2018,

representing 13 ENGS-affected chimpanzees (cases) and 17 apparently healthy chimpanzees (controls).

Response: We thank the reviewer for the observant reading of the manuscript. We have carefully examined the data and found an inconsistency with how we reported the total number (“analyses” and “examinations” were defined differently) and a switch in the Methods section of the numbers 13 and 17 regarding cases and controls. We have rectified the inconsistency by reporting “analyses” throughout (lines 149, 566) and have fixed the typo (line 567).

Minor comments:

Comment 7: *Suppl. Tab 1: It would greatly enhance the readability if information in column “Signs” was stated in clear text instead of the number code. The same is true for the abbreviations for the tissues. Maybe the authors could use landscape format for this table to gain enough space for this information. Please also consider including information on the samples used for the different analyses (see above).*

Response: We agree that Supplemental Table 1 would benefit from re-formatting. We modified the table to increase readability as suggested by converting it to landscape format and using text labels instead of a number code (now Supplementary Table 2). Additionally, we added a “Comprehensive Testing Results” table to clearly show which samples were used for each assay (Supplementary Table 3; see response to Reviewer 1, above).

Comment 8: *Suppl. Tab. 2: “Fungi” is no genus; numbers in brackets are no percentages as stated in footnote a.*

Response: Good catch! We converted the values in brackets to percentages and removed the line for fungi since it is not relevant to the focus of the table, which is parasites. We then moved this information into a new table in which we consolidated statistical analyses from across pathogen types (Supplementary Table 5; see response to Reviewer 1, above), such that the new table replaces this table.

Comment 9: *Spelling of “Ole” in Suppl. Fig 2 and Suppl. Tab. 1 differs (“Ole” vs. “Olé”).*

Response: Another good catch. We corrected the spelling to “Olé” (Supplementary Figure 2).

Comment 10: *Please check for “gastric dilation” and “gastric dilatation” (for example, in the sentence in lines 370 – 373 both appear to be used synonymously).*

Response: Great point. We have determined that “gastric dilation” is the most accurate terminology and have corrected the text accordingly (lines 125, 274, 418, 420).

Comment 11: *In line 443 a closing “)” is missing after “MIC = 0.50 mg/l.”*

Response: Many thanks. We have fixed the typo (line 497).

Reviewer #3 (Remarks to the Author):

Comment 1: *The manuscript describes the identification of a new bacterial species associated with acute to peracute death in chimpanzees. The manuscript will likely be of interest to the readership.*

Response: Many thanks for the positive appraisal.

Comment 2: *The main challenges in this manuscript relate to the definition of cases, lack of data points for many cases, and the inconsistent sampling. Much of this may be due to the sanctuary setting and problems associated with opportunistic sampling.*

Response: We thank the reviewer for this insightful critique. We agree that more consistent sampling and records would improve the strength of the study. The reviewer is indeed correct that these challenges largely stem from the setting, and ultimately from resource availability. To address this point, we have added a section to the Discussion clarifying this issue (lines 457-463).

Comment 3: *Case definition: The 56 cases that comprise “epidemic neurologic and gastroenteric syndrome” (ENGS) appear to be of two distinct types. About 60% (32 cases) have varied neurologic and gastric symptoms; the other 40% (24 cases) appear to be sudden death or found dead animals, without prior signs. Although several clostridial organisms can do this, initially combining these into one syndrome (before the bacterial findings) seems a bit tenuous.*

Response: We agree that the syndrome can appear tenuous. The manuscript may not have adequately captured the fact that this syndrome is indeed a syndrome, in that clinical signs are characteristic and always associated. We have therefore added text to the Results, Epizootiology, Clinical Signs, and Pathology Section clarifying this point (lines 112-114) and also added a statement to the Discussion acknowledging that the case definition will likely become more refined as the syndrome is studied more (lines 364-365).

Comment 4: *Was ENGS initially identified as both symptomatic and sudden death animals, or did this realization occur over time?*

Response: This is an interesting question. ENGS was identified as both symptomatic and sudden death animals, and it did not take time to realize that it was a distinct disease. Veterinarians involved in the cases describe the syndrome as “unmistakable,” “horrendous,” “awful,” and other similar adjectives indicating how unmistakable it is. We have therefore added a statement in the Results, Epizootiology, Clinical Signs, and Pathology section clarifying the lack of ambiguity about the syndrome and that it was immediately identifiable (lines 111-112).

Comment 5: *Post mortem evaluation: Post mortem evaluations were only available for 19 animals, with only 14 of these having the gastrointestinal tract examined, and only 11 with gastric distention. It would seem that this would represent a much better-defined group to initially focus on for the various assays. After finding the bacteria, then the thought would occur to evaluate the sudden death cases. The authors could then introduce the findings in the sudden death cases to show the bacteria was also present in these cases and suggest that they are variations in presentation. Logically, that is how I would have assumed this process to have occurred.*

Response: This is very insightful; we had not considered the possibility of organizing the flow of the findings this way. Because of the remote setting and resource limitations, not all post-mortem evaluations included sample collection, so we unfortunately do not have samples for all of these cases. To clarify the relationship between individual cases, post-mortem examination records, clinical presentation, and sample collection, we have created a new summary graphic (Figure 1; see response to Reviewer 2, above) and added clinical presentation information to the table detailing the samples used in the study (Supplementary Table 2). Furthermore, we have added text in the Discussion to address the data gaps and overall study caveats associated with opportunistic sampling (lines 457-462).

Comment 6: *Samples: Based on Supplementary Table 1, actual tissue samples were only obtained from 7 cases and 1 control. All other samples were blood or body fluids. It would be best to use GI tissues, assuming they are available.*

Response: We are grateful to the reviewer for making this excellent suggestion. Unfortunately, the included samples were the only available materials suitable for our analyses, though we hope to collect more GI tissues from future cases for further study. To clarify this point, we have added text to the Discussion to this effect (lines 459-461).

Comment 7: *Samples II: It is hard to assimilate which of the various assays were performed for the various samples. Were all samples evaluated for all assays (Parasitology, virology, etc?). I am pretty certain they were not, however a table listing all the samples and which ones were tested by each assay (and which were positive) would greatly improve interpretation b the reader.*

Response: We agree with the reviewer. To address this point, we have compiled the test records from each sample and organized them into a table by individual and sample type. The resulting “Comprehensive Testing Results” table was added to the supplemental information (Supplementary Table 3; see response to Reviewer 1, above).

Comment 8: *In the virology section, it states “no viruses were case-associated”. I am assuming that means that no viruses were found in “syndrome” cases, just in controls? It was unclear, at least to me.*

Response: The reviewer raises an important point which indeed requires clarification. To address this issue, upon the first use in the text, we have clearly defined “case-associated” by modifying the text in the Results, Parasitology section (lines 172-173). Furthermore, we have created a “Case/Control Statistical Summary” table of results to demonstrate the statistical definition of “case-associated” for each assay (Supplementary Table 5; see response to Reviewer 1, above).

Comment 9: *I strongly agree with the authors (lines 373-376) that “A more consistent gross and histopathologic evaluation of affected chimpanzees in the future” is needed to better define the syndrome. However, I think that can be addressed in later studies.*

Response: We agree, and we thank the reviewer for acknowledging that there is a good amount of work yet to be done! It is indeed a top priority to address the caveats of this study and expand the characterization of ENGS in follow-up studies, and we are working with the veterinarians in Sierra Leone on next steps.

Comment 10: *Although the above may sound negative, overall, I feel the authors have done a good job in trying to present a hard to define syndrome using the materials and data available. My major issues involve addressing more clearly the areas above to more accurately describe the various caveats and gaps in the data. I find their overall premise and reasoning (and the data they have) consistent with their claims.*

Response: Many thanks for the positive comment! We truly appreciate the reviewer’s encouragement and careful critique of our manuscript.

REVIEWERS' COMMENTS

Reviewer #1 (Remarks to the Author):

The authors addressed the comments that I raised previously.
I have no further remarks

Reviewer #2 (Remarks to the Author):

All points I raised were sufficiently addressed and clarified. Thereby, the clarity of the manuscript and the possibility to follow the train of thought of the authors have been improved.
I have only one additional minor point: The data presented in the newly added supplementary table 1 need a clarification. Likely, low and high refer to min and max values or are these numbers derived from boxplot statistics and therefore outliers were removed? In other words, the origin of the given numbers needs to be clarified.

Reviewer #3 (Remarks to the Author):

The manuscript describes the identification of a new bacterial species associated with acute to peracute death in chimpanzees. The manuscript will be of interest to the readership.

There remain significant challenges in this manuscript relating to the definition of cases, lack of data points for many cases, and the inconsistent sampling. As acknowledged by the authors, much of this is due to the sanctuary setting and problems associated with opportunistic sampling. I believe that the authors have addressed these issues where possible.

The authors have meaningfully addressed the issues of case definition – primarily based on their observations that these cases were recognized as all related (a syndrome) due to similar gross and histologic findings, before the analysis began.

The lack of data points and inconsistent sampling is expected and I was unsurprised that little could be done to add to those; again, due to the remote sanctuary setting. The authors readily acknowledge this issue and have stated so in the manuscript. I am encouraged that they have a sampling plan going forward to better document future cases.

I believe that the significant amount of added materials clarifies and presents the data more completely and raises significantly less questions. As far as the clinical and pathological signs and grouping these cases together as a single syndrome, I think they have done the best possible with the information at hand.

The work involved in this manuscript is extensive, and if the other reviewers are satisfied with the molecular and microbiological details, I recommend approval of the submission.

“A novel bacterium in the genus *Sarcina* linked to a uniformly lethal epizootic disease in sanctuary chimpanzees (*Pan troglodytes verus*) in Sierra Leone”

For publication in *Nature Communications*

Point-by-point responses to reviewer comments

Reviewer #1 (Remarks to the Author):

Comment 1: *The authors addressed the comments that I raised previously. I have no further remarks*

Response 1: Many thanks for the positive comment and for the careful comments about our initial submission.

Reviewer #2 (Remarks to the Author):

Comment 1: *All points I raised were sufficiently addressed and clarified. Thereby, the clarity of the manuscript and the possibility to follow the train of thought of the authors have been improved.*

Response 1: Thank you! We appreciate the positive assessment.

Comment 2: *I have only one additional minor point: The data presented in the newly added supplementary table 1 need a clarification. Likely, low and high refer to min and max values or are these numbers derived from boxplot statistics and therefore outliers were removed? In other words, the origin of the given numbers needs to be clarified.*

Response 2: We agree; the values in the table need to be clarified. We have therefore modified supplementary table 1 by using the more appropriate labels “Min” and “Max” and added a footnote to supplementary table 1 stating that “values were calculated using records compiled from all ENGS deaths that occurred from 2005 through 2018, with 56 cases in total.”

Reviewer #3 (Remarks to the Author):

Comments 1-6:

- *The manuscript describes the identification of a new bacterial species associated with acute to peracute death in chimpanzees. The manuscript will be of interest to the readership.*
- *There remain significant challenges in this manuscript relating to the definition of cases, lack of data points for many cases, and the inconsistent sampling. As acknowledged by the authors, much of this is due to the sanctuary setting and problems associated with opportunistic sampling. I believe that the authors have addressed these issues where possible.*

- *The authors have meaningfully addressed the issues of case definition – primarily based on their observations that these cases were recognized as all related (a syndrome) due to similar gross and histologic findings, before the analysis began.*
- *The lack of data points and inconsistent sampling is expected and I was unsurprised that little could be done to add to those; again, due to the remote sanctuary setting. The authors readily acknowledge this issue and have stated so in the manuscript. I am encouraged that they have a sampling plan going forward to better document future cases.*
- *I believe that the significant amount of added materials clarifies and presents the data more completely and raises significantly less questions. As far as the clinical and pathological signs and grouping these cases together as a single syndrome, I think they have done the best possible with the information at hand.*
- *The work involved in this manuscript is extensive, and if the other reviewers are satisfied with the molecular and microbiological details, I recommend approval of the submission.*

Response 1-6: We appreciate the reviewer's positive comments on the manuscript and suggestions for future work. We are indeed pursuing this topic and, building on our current results, are striving to fill additional knowledge gaps.